# Fresh groundwater discharge insignificant for the world's oceans but important for coastal ecosystems

Elco Luijendijk [1✉], Tom Gleeson [2] & Nils Moosdorf [3,4]

The flow of fresh groundwater may provide substantial inputs of nutrients and solutes to the oceans. However, the extent to which hydrogeological parameters control groundwater flow to the world's oceans has not been quantified systematically. Here we present a spatially resolved global model of coastal groundwater discharge to show that the contribution of fresh groundwater accounts for ~0.6% (0.004%–1.3%) of the total freshwater input and ~2% (0.003%–7.7%) of the solute input for carbon, nitrogen, silica and strontium. However, the coastal discharge of fresh groundwater and nutrients displays a high spatial variability and for an estimated 26% (0.4%–39%) of the world's estuaries, 17% (0.3%–31%) of the salt marshes and 14% (0.1–26%) of the coral reefs, the flux of terrestrial groundwater exceeds 25% of the river flux and poses a risk for pollution and eutrophication.

[1] Geoscience center, University of Göttingen, Goldschmidstrasse 3, 37077 Göttingen, Germany. [2] Department of Civil Engineering and School of Earth and Ocean Sciences, University of Victoria, 3800 Finnerty Road, Victoria, BC V8P 5C2, Canada. [3] Leibniz Centre for Tropical Marine Research (ZMT), Fahrenheitstr. 6, 28359 Bremen, Germany. [4] Institute of Geosciences, Kiel University, Ludewig-Meyn-Straße 10, 24118 Kiel, Germany. ✉email: elco.luijendijk@geo.uni-goettingen.de

Submarine groundwater discharge (SGD), the flow of fresh or saline groundwater to oceans, may be an important contributor to the water and chemical budgets of the world's oceans[1,2]. The fresh component of submarine groundwater discharge is critical, due to its high solute and nutrient loads[3], it has been estimated to be up to 10% of the river discharge to the world's oceans[1] and to equal the inputs by rivers for solutes such as carbon[4], iron[5], silica[6,7], and strontium[8]. In addition, fresh SGD and potentially buffers ocean acidification with groundwater alkalinity[9]. The total flux of groundwater to the ocean can be divided into three distinct fluxes: fresh SGD, near-shore terrestrial groundwater discharge (NGD) and recirculated sea water. We consider to be meteoric groundwater discharging below the mean sea level to constitute fresh SGD, whereas meteoric groundwater discharging above the mean sea level near the coastline is termed NGD. Fresh SGD and NGD are driven by recharge from onshore precipitation and are critical due to their high solute and nutrient loads whereas recirculated sea water is driven by mechanisms such as waves, tides, storm surges and density-dependent flow. Fresh SGD and NGD combine to form coastal groundwater discharge (CGD). Note that we use the long-term (annual) mean sea level to separate SGD and NGD, and not the high tide line as used by some previous studies.

Although a large number of studies have estimated fresh SGD locally, it is difficult to derive a global estimate from local measurements of fresh SGD, because they are highly variable, often uncertain and strongly biased towards high submarine groundwater discharges[1]. Global-scale estimates of fresh SGD vary by four orders of magnitude, and range up to 10% of the river discharge towards the oceans[1,2,10]. A recent estimate for the USA[11] and a near-global estimate[12] suggest that fresh SGD is at the lower end of most earlier estimates. However, these have been based on surface water budget calculations only, and their consistency with groundwater flow processes in the subsurface is uncertain. In addition, previous large-scale estimates have not reported the near-shore terrestrial discharge of groundwater that may affect coastal water and solute budgets, evapotranspiration and ecosystems, even though the existence of onshore groundwater seepage near surface water features has long been recognized for lakes, wetland, or streams[13,14].

Here, we use numerical models of density-dependent groundwater flow to quantify the partitioning of terrestrial and submarine groundwater discharge and the sensitivity of coastal groundwater discharge to controlling variables such as topography, permeability, recharge and size of contributing area. We subsequently quantify coastal groundwater discharge at the global scale by combining a series of model experiments with a global geospatial analysis of controlling variables. The results highlight that for the majority of the world's coastline coastal groundwater discharge is limited by the flow capacity of the subsurface and not by the volume of groundwater recharged. The results also show that the global flux of groundwater and solutes to the oceans is low, but that the high variability of groundwater discharge results in locally important fluxes to coastal ecosystems. Our study is a significant advance on recently published large-scale estimates of fresh SGD[11,12] that are based on surface water budgets of coastal watersheds, because it explicitly takes into account groundwater flow processes in the subsurface to resolve all three fluxes of groundwater to the ocean, including density-dependent flow that is critical for resolving submarine and terrestrial coastal groundwater fluxes, and because we use the best available global distributed input data.

## Results and discussion
**Controls on coastal groundwater discharge**. We modeled coastal groundwater flow using a numerical model in which groundwater

discharges both on land and at the seafloor and the location and rate of discharge are calculated by the model (see Methods section for a detailed description of the model approach). The model experiments are based on a conceptual model shown in Fig. 1a. The modeled groundwater flow paths, salinity and discharge rates for a typical model setup are shown in Fig. 1b. The results show that offshore freshwater discharge is mirrored by a zone of near-shore terrestrial groundwater discharge (NGD) (Fig. 1b). A model sensitivity analysis that explores the response of groundwater discharge to variation in groundwater recharge, size of contributing area, permeability and topographic gradient shows that NGD is higher than fresh SGD in most settings except at high permeability or topographic gradients (Fig. 1d–g). NGD peaks at the coastline and decays exponentially with distance to the coast (Fig. 1b and Supplementary Fig. 1). The modeled NGD flux represents a mixture of evapotranspiration, ponding, surface runoff and lateral groundwater flow perpendicular to the cross-section towards streams. Note that in our model experiments large areas show a zero net flux across the land surface (Fig. 1b and Supplementary Fig. 1), where groundwater recharge and discharge are equal, in contrast to the focused and relatively high discharge flux near the coastline.

Model sensitivity analysis demonstrates that coastal groundwater discharge is predominantly controlled by the flow capacity of coastal aquifers, which is the product of the topographic gradient, the permeability and thickness of coastal aquifers. The topographic gradient governs the maximum hydraulic gradient that can be attained, which in turn controls the groundwater that can flow towards the coastline; the higher the gradient, the higher the coastal groundwater discharge (Fig. 1e). The topographic gradient also governs the partitioning of submarine and terrestrial coastal discharge. At low topographic gradients fresh SGD is lower than NGD, because of the extra force that is required to displace denser and more saline groundwater offshore to generate fresh SGD. However, at high topographic gradients the watertable is below the surface for most of the terrestrial parts of coastal groundwater systems, except for areas that are close to the coastline. Because terrestrial groundwater discharge only takes place where the watertable is at the surface, NGD is relatively low in such settings and most of the coastal groundwater contributes to fresh SGD (Fig. 1e).

The permeability of coastal aquifers exerts a strong control on the amount of groundwater that can be driven through the system towards the coast and the magnitude of coastal groundwater discharge (Fig. 1g). Permeability also controls the hydraulic gradient, the higher the permeability, the lower the gradient that is required to channel the same amount of groundwater to the coast. Therefore, for high values of permeability coastal discharge shifts away from NGD towards fresh SGD. While in principle the thickness of aquifers is equally important as permeability, the variation in permeability in our global analysis of coastal aquifers is 10 orders of magnitude (Fig. 1g and Supplementary Fig. 2), whereas the thickness of coastal aquifers is expected to vary one to two orders of magnitude following previous studies on the variation of the depth of modern groundwater in the subsurface[15,16], which is a proxy for the part of the subsurface where the majority of meteoric water is channeled through.

Somewhat counterintuitively, groundwater recharge and size of the contributing area do not influence coastal groundwater discharge in most cases. At low values of groundwater recharge or contributing area the recharge volume is below the flow capacity of coastal aquifers, and increases in recharge rate or contributing area result in an increase in coastal groundwater discharge. However, once a threshold value is reached where the recharge volume equals the flow capacity of the subsurface any further increase in recharge volume only generates additional terrestrial

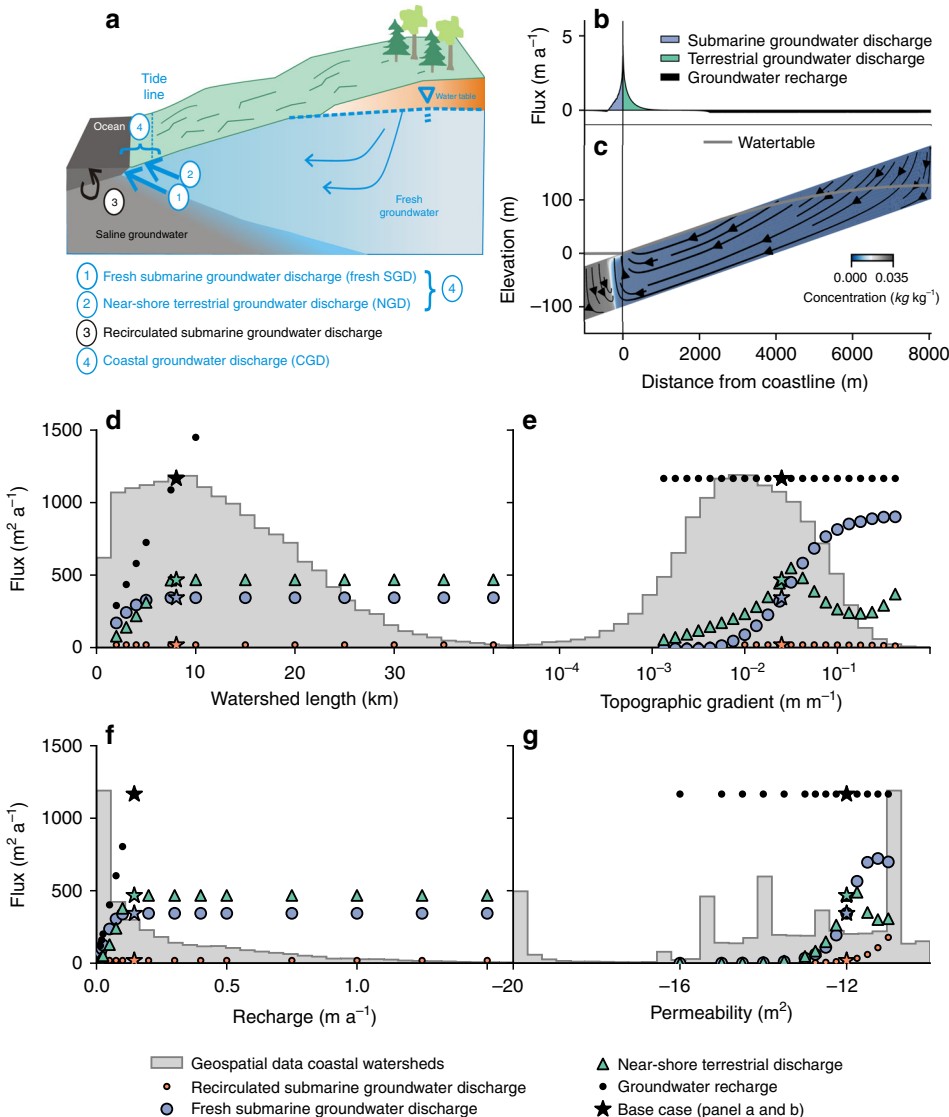

**Fig. 1 Modeled sensitivity of coastal groundwater discharge to hydrogeological parameters.** The model results demonstrate that the flux of groundwater to the ocean is controlled primarily by topographic gradient (**e**) and aquifer permeability (**g**) and is relatively insensitive to watershed length (**d**) and groundwater recharge (**f**). The conceptual model that the sensitivity analysis is based on is shown in **a**. **b**, **c** Modeled groundwater fluxes over the land surface and seabed (**b**), groundwater flowlines and salinity (**c**) for a base-case model run that represents a coastal watershed with global median values for groundwater recharge (0.143 m a$^{-1}$), but with relatively high permeability (10$^{-12}$ m$^2$) and topographic gradient (2.5%). Each dot in panels **d**–**g** represents the modeled coastal groundwater recharge or discharge fluxes for a single model run. The discharge fluxes are subdivided into modeled fresh submarine groundwater discharge, near-shore terrestrial discharge and recirculated submarine groundwater discharge as described in the main text. The histograms in panels **d**–**f** show the distribution of the controlling parameters in all 40,082 global coastal watersheds. The results of the model sensitivity analysis are available as Supplementary Data 1.

discharge away from the coast, whereas coastal groundwater discharge and its two components (fresh SGD and NGD) stay constant (Fig. 1d, f). In our sensitivity analysis this threshold is reached well below values of recharge and contributing area that correspond to the median coastal aquifer in our global analysis. This is in spite of the use of relatively high values for permeability and topographic gradient as a base case for the sensitivity analysis (Fig. 1d, f). Therefore, for most coastal groundwater systems globally the bottleneck for coastal groundwater discharge is expected to be their flow capacity and not the volume of recharge that is added to these systems. This agrees with results by Michael et al.[17] that groundwater flow in the majority of the world's coastline is topography-limited instead of recharge-limited.

Comparison of the modeled CGD with the distribution of the controlling parameters in coastal aquifers shows that CGD is an insignificant part of the total groundwater flux for most coastal groundwater systems and most groundwater discharges on land and contributes to surface runoff, river baseflow, and evapo-transpiration before it reaches the coast (Fig. 1d–g). For a separate model run that used the same parameters as the base case of the model sensitivity analysis, but with the global median values of permeability and topographic gradient as derived from a global geospatial analysis of coastal watersheds (see Methods section), modeled CGD is only 2.0% and the modeled fresh SGD is only 0.5% of the total recharge volume. The discharge flux reaches a maximum of 0.23 m a$^{-1}$ at the coastline (Supplementary Fig. 1), which is lower than the lowest measured value of fresh SGD in the literature known to us[18]. This signifies that in the majority of the world's coastline fresh SGD is expected to be so low that it is difficult to measure using techniques like seepage meters. For the

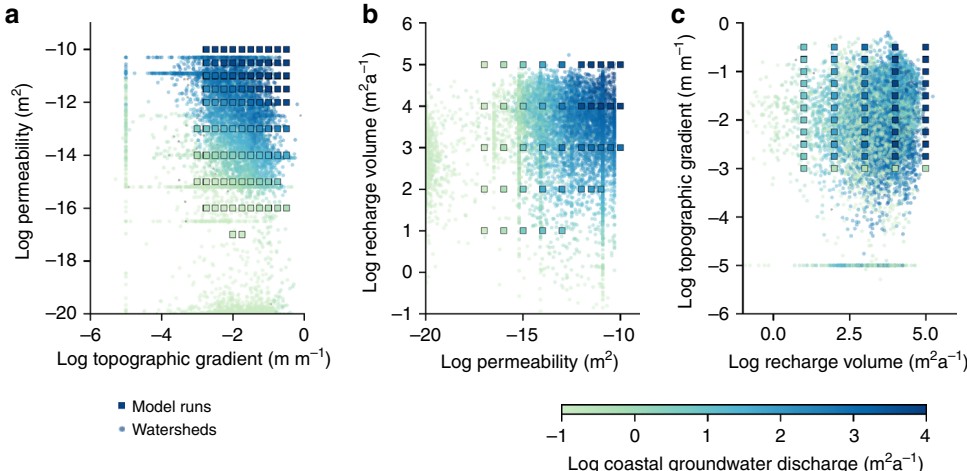

**Fig. 2 Illustration of the linear interpolation method used to quantify coastal groundwater discharge at a global scale.** This figure shows the interpolation of coastal groundwater discharge for each of the 40,082 global coastal watersheds from a series of 351 model runs that cover the range of permeability, topographic gradients and groundwater recharge found in these watersheds. Each dot represents the parameter values for permeability, topographic gradient and recharge volume for a single coastal watershed, which were derived from a global geospatial analysis (see Methods section). The watersheds are colored by their interpolated value of coastal groundwater discharge. Each square represents a single numerical model run. Each panel shows the parameter values and interpolated CGD for a pair of parameters; permeability and topographic gradient (**a**), permeability and recharge volume (**b**) and recharge volume and topographic gradient (**c**). Note that in each panel there are a number of model runs (squares) that overlap. In each case the model run is shown with the highest modeled coastal groundwater discharge. For instance, for panel **a** behind each square there are a number of additional model runs that are not shown that used different recharge volume values. The results of the model experiments can be found in Supplementary Data 2. The geospatial data of coastal watersheds and the interpolated values of coastal groundwater discharge are available as Supplementary Data 3.

base case model run in our sensitivity analysis, which represents a coastal groundwater system consisting of relatively permeable rocks and a relatively high topographic gradient of 2.5% (Fig. 1b), 50% of the recharge volume contributes to coastal groundwater discharge. While coastal groundwater discharge is insignificant in most cases, when permeability exceeds a threshold value of $10^{-12}$ m$^2$ and topographic gradients exceed 1%, coastal groundwater discharge increases rapidly and can become the dominant groundwater discharge component (Fig. 1e, g). In such locations, coastal groundwater discharge can strongly influence coastal ecosystems, as shown in local studies[19], and could act as freshwater resource for the coastal population[20].

**Global coastal groundwater discharge.** Due to the relatively small area of coastal watersheds compared to the watersheds feeding the world's rivers, even if all groundwater recharge in coastal watersheds were to discharge directly into the oceans, CGD would not exceed 5.5% of the river input into the world's oceans. However, as discussed in the previous section, for most coastal watersheds CGD is only a fraction of the total recharge volume.

Global CGD, fresh SGD, and NGD were calculated by linear interpolation of the results of 351 model runs to geospatial data of 40,082 coastal watersheds (Fig. 2 and Supplementary Figs. 3, 4). For each of the world's coastal watersheds the CGD, NGD, and fresh SGD fluxes were calculated by linear interpolation of the CGD, NGD, and fresh SGD fluxes in the model runs with the values of permeability, recharge volume and topographic gradient that were closest to the values of each watershed (see Methods section). The results show that CGD is an insignificant contributor to the water budget of the world's oceans (Fig. 3), and is equal to 224 (1.4–500) km$^3$ a$^{-1}$, which is 13 (0.2–26)% of the groundwater recharged in coastal watersheds. The calculated global CGD equals 0.6% (0.004–1.3)% of the river discharge to the oceans[21]. Of the total CGD, an estimated 147 (1.0–290) km$^3$ a$^{-1}$ discharges onshore as NGD and 78 (0.4–210) km$^3$ a$^{-1}$ discharges offshore as

fresh SGD (Supplementary Fig. 5). This is 0.4 (0.002–0.8)% and 0.2 (0.001–0.6)% of the river discharge, respectively, which is at the lower end of previous global estimates[1,10]. The uncertainty ranges reported here represent the CGD, NGD, and fresh SGD fluxes calculated using end-member estimates for permeability, groundwater recharge, contributing area and topographic gradient. The uncertainty of global CGD is mostly caused by the high uncertainty of the values of permeability that were used[22], which is on average 2 orders of magnitude (see Supplementary Fig. 2). Additional sources of uncertainty are the representative topographic gradient of coastal watersheds, groundwater recharge, and the size the area that contributes to CGD. CGD would be 164 km$^3$ a$^{-1}$ instead of 224 km$^3$ a$^{-1}$, when a lower estimate of topographic gradient would be adopted that follows the average gradient of coastal streams instead of the average gradient of entire watersheds. On average the difference between the two estimates of topographic gradient is 0.4%. Two alternative global models of groundwater recharge[23,24] with recharge rates that differ on average 50% result in an uncertainty range of 164–224 km$^3$ a$^{-1}$. If the contributing area is assumed to be twice the size of coastal watersheds the calculated CGD is 329 km$^3$ a$^{-1}$. While the uncertainty ranges reported here are large, the best estimates of permeability, topographic gradient and recharge provide a good fit of the model to observed watertable gradients in coastal watersheds (Supplementary Fig. 6), which suggests that the reported best estimates of CGD are relatively robust. Note that model experiments demonstrate that the partitioning of coastal groundwater discharge in submarine and terrestrial discharge is highly sensitive to the local topographic gradient (Fig. 1d), and therefore the total coastal groundwater discharge is a more robust estimate than the onshore (NGD) and offshore (fresh SGD) components.

The calculated fresh SGD is only a minor fraction, 0.06% (0.0003%–0.2%), of the global total SGD flux, which includes recirculated seawater, and has been estimated as three to four times of the river flux globally based on measured concentrations of radiogenic radon in seawater[25]. This is in line with analytical models that estimated seawater circulation due to tidal and wave

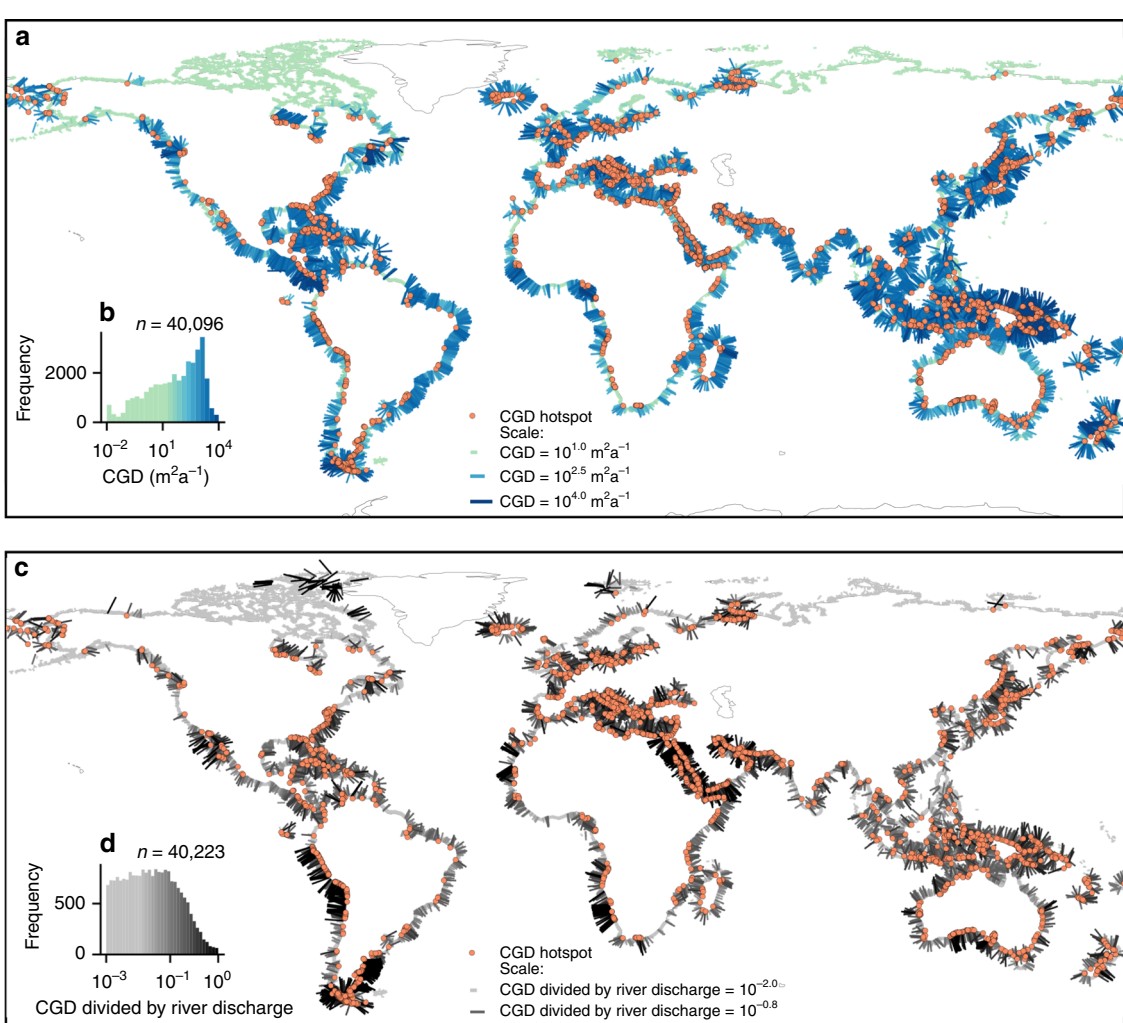

**Fig. 3 Global maps of coastal groundwater discharge and comparison with surface water discharge.** The maps show that the flow of groundwater to the oceans is highly variable, dominated by localized hotspots, and can locally be a source of water that rivals surface water discharge. **a** Coastal groundwater discharge (CGD), and **b** CGD divided by the surface water flux to the oceans. The coastal groundwater discharge and surface water discharge data that are shown here are available in Supplementary Data 3.

forcing to be roughly equal to the estimated total SGD flux[26]. The very low terrestrial contribution to the overall SGD flux means that global SGD consists almost exclusively of recirculated seawater and the net-input of solutes to the oceans by SGD is much lower than previously assumed. A large part of the total CGD flux discharges on land as NGD, which takes place in a zone that on average extends 400 m from the shore (Supplementary Fig. 7). In most cases NGD does not exceed potential evapotranspiration rates. However, NGD exceeds potential evapotranspiration[27] in 28 (0.07–58)% of the global coastline, where it contributes to surface runoff and baseflow close to the shoreline.

**Transport of solutes to the oceans.** A first order estimate of the transport of carbon, nitrogen, silica, and strontium to the world's oceans based on published compilations of the average solute concentrations in coastal groundwater[7,8,28,29] suggest the contribution of CGD is ~2% of the input by rivers (Table 1), which is much smaller than most earlier estimates[30], including the up to 100% contribution suggested by some recent studies that extrapolated global inputs from local and regional-scale estimates[4–6]. The difference is most likely the result of bias in reporting of fresh SGD, scaling up high local rates of fresh SGD, and the difficulty of separating fresh and recirculated SGD in measurements. There

are insufficient data available for the concentration of iron in coastal groundwater, but even with relatively high concentrations of 40 mg L$^{-1}$ that exceed local estimates in the literature[5] the contribution would equal 2% of the river flux. Note that solute transport by onshore discharge (NGD) component of CGD is uncertain. Much of NGD will be transpired, but solutes may eventually still be transported to the oceans at high tide or flood events. The estimates decrease to 1% when only fresh SGD is assumed to contribute to the solute transport to the oceans instead of the total CGD. Our estimate of solute fluxes to the oceans by CGD is a first order, spatially aggregated average that assumes conservative transport. The spatially distributed map of coastal groundwater discharge (Fig. 3) provides the possibility of better future estimates for locations where concentrations of specific solutes are known.

**Hotspots of coastal groundwater discharge.** Although the overall contribution of water and solutes by coastal groundwater discharge to the oceans is low, coastal groundwater discharge is highly variable, with 10% of the global coastline contributing 90% of the total discharge. Comparison with data on the discharge of rivers to the oceans[23] shows that coastal groundwater discharge volume can locally be close to the river input (Fig. 3a, b),

**Table 1 Comparison of published and new values of the calculated solute flux to the oceans by rivers and coastal groundwater discharge.**

| Solute | River flux | Previous estimate of solute flux by fresh SGD | Average concentration in CGD | New estimate of solute flux by CGD | | References |
|---|---|---|---|---|---|---|
| | (kg a$^{-1}$) | (% of river flux) | (mg L$^{-1}$) | (kg a$^{-1}$) | (% of river flux) | |
| Dissolved inorganic carbon | 7.1 (6.5-7.7) × 10$^{11}$ | 23% (17%-39%) | 60 | 1.3 (0.01-3.0) × 10$^{10}$ | 1.9% (0.011-4.6%) | 83 |
| Dissolved inorganic nitrogen | 1.9 × 10$^{10}$ | 7.5% | 2 | 4.5 (0.03-10) × 10$^{8}$ | 2.4% (0.015-5.3%) | 28-30,89 |
| Dissolved silica | 1.7 × 10$^{11}$ | 8% | 8 (6-11) | 2.4 (0.01-2.4) × 10$^{9}$ | 1.4% (0.003-5.1%) | 7,90 |
| Strontium | 2.9 (1.6-4.1) × 10$^{9}$ | 21% (15-149%) | 0.25 | 5.7 (0.04-13) × 10$^{7}$ | 2.0% (0.009-7.7%) | 8 |

The comparison shows that the contribution of fresh groundwater is much lower than previously assumed. The calculated solute fluxes are based on the modeled coastal groundwater discharge (CGD) reported here and previously reported values of the average concentrations of solutes in coastal groundwater systems that are referenced below and discussed in the Methods section. Note that all previous estimates of the solute flux cited here are based on earlier estimates of fresh submarine groundwater discharge (SGD) instead of CGD. The brackets denote minimum and maximum estimates, which are based on reported values in the literature for the river flux, previous estimate fresh SGD or the average concentration in CGD columns, and on the minimum and maximum estimates of CGD as discussed in this manuscript.

which given its relatively high solute load[3] means that it may dominate the solute input in some coastal ecosystems. We define coastal groundwater discharge hotspots as watersheds where the coastal groundwater discharge exceeds 100 m$^2$ a$^{-1}$ and 25% of the river discharge. The threshold value of 100 m$^2$ a$^{-1}$ reflects a relatively conservative lower bound for reported values at locations with high coastal groundwater discharge and associated ecosystem impacts (see Methods section and Supplementary Tables 1, 2). Coastal groundwater discharge hotspots (Fig. 3a, b) cover 9 (0.02–30)% of the global coastline and are predominantly located in areas with a steep coastal topography due to glacio-isostatic rebound, active tectonics or volcanic activity and areas consisting of permeable unconsolidated sediments, carbonates or volcanic rocks (Supplementary Fig. 8). The distribution of hotspots is consistent with documented sites of high fresh groundwater discharge globally that are predominantly located in North America, Europe, and East Asia (Supplementary Tables 1, 2). However, at many hotspots, such as Iceland and parts of South America, Africa, and South Asia, and many tropical islands coastal groundwater discharge has been unexplored to our knowledge.

**Local impacts of coastal groundwater discharge.** Due to its high spatial variability, fresh groundwater discharge can locally have strong impacts on coastal hydrology and ecosystems. Coastal groundwater discharge can control the salinity, nutrient budget and productivity of coastal lagoons[31], salt marshes[32], and mangroves[33], and the associated solute flux can cause eutrophication[34], algal blooms[35], and the degradation of coral reefs[36], but it can also increase fish fitness[37]. In particular, tropical islands consisting of volcanic and carbonate rocks are likely to host high fresh groundwater fluxes (Supplementary Fig. 8) that can supply crucial nutrients to marine ecosystems that may be located far from other nutrient sources[38]. A first order estimate of global coastal groundwater eutrophication risk (Fig. 4a) shows that 13% (0.2–23%) of the global coastline is at risk of eutrophication by terrestrially derived groundwater and nutrients. Eutrophication risk is defined by nitrogen application in coastal watersheds that exceeds 10 kg ha$^{-1}$ and coastal groundwater discharge that exceeds 100 m$^2$ a$^{-1}$. The threshold value for nitrogen input corresponds to values that have historically led to strongly elevated nitrogen concentrations in groundwater in Europe and North America[39,40] that have contributed to the eutrophication of terrestrial and nearshore ecosystems[41]. Note that the lack of data outside of North America and Europe make the extrapolation uncertain, and the numbers reported here should be considered a first order estimate. Coastal areas with elevated risk include sensitive coastal ecosystems. A comparison with published locations of these ecosystems[42–44] shows that 26% (0.4–39%) of the world's estuaries and 17% (0.3–31%) of the salt marshes at risk of eutrophication. In addition, 14% (0.1–26%) of the coastline that is

located within 500 m of a coral reef is at risk of eutrophication (Fig. 4a). A review of sites with documented ecosystem impacts of coastal groundwater discharge (Supplementary Table 2) suggests that the threshold values of coastal groundwater discharge and nitrogen input in the adjacent coastal watersheds that we used to define high risks are relatively conservative and that adverse ecosystem impacts may also occur at lower threshold values for nitrogen application or coastal groundwater discharge. In addition to its importance for coastal ecosystems, coastal groundwater discharge can locally also be a freshwater resource that is used as drinking water or for other purposes in a limited number of locations, but has been generally overlooked[20]. However, this resource may also be sensitive to pollution, and exploitation of this resource would need to be carefully managed to avoid salt-water intrusion and adverse impacts on coastal ecosystems.

The values of global coastal groundwater discharge reported here represent a natural undisturbed system and do not include groundwater pumping. However, many coastal groundwater systems may be affected by groundwater pumping. A comparison of the calculated coastal groundwater flux and published spatially resolved values of average annual groundwater depletion in coastal watersheds[45] shows that in most coastal groundwater systems are not associated with depletion (Fig. 4b). The global rate of groundwater depletion in coastal watersheds equals 9.2 km$^3$ a$^{-1}$, which is much lower than the coastal groundwater discharge. However, groundwater depletion is highly localized and while in part of the coast depletion is likely buffered by a decrease in coastal groundwater discharge, in 13% (8–19%) of the global coastal watersheds groundwater depletion exceeds coastal groundwater discharge. The depletion in these watersheds exceeds CGD by 6.7 (5.3–8.5) km$^3$ a$^{-1}$. In these watersheds coastal groundwater discharge has already or will reduce to zero at some stage in the future and instead seawater will start intruding into terrestrial groundwater systems.

**Comparison with large-scale estimates of groundwater flow.** Our model results estimate a coastal groundwater discharge flux for the contiguous US of 8.5 (0.1–17) km$^3$ a$^{-1}$, which is in the same range as the 15 ± 4 km$^3$ a$^{-1}$ estimated by Sawyer et al.[11]. A recently published near-global estimate[12] yielded a fresh SGD flux of 489 ± 337 km$^3$ a$^{-1}$, which is likely an overestimate due to the relatively coarse watershed database that this estimate relies on[12,46]. This is higher than the CGD flux of 224 (1–500) km$^3$ a$^{-1}$ and the fresh SGD flux of 78 (0.4–210) km$^3$ a$^{-1}$ reported here. However, these two studies cannot be compared directly with our results, because they are based on the assumption that groundwater discharge is controlled solely by surface morphology and drainage density and that all groundwater in watersheds without mapped streams discharges as fresh SGD. This assumption contrasts with our model experiments, which highlight the role of the flow capacity of the subsurface and the key role of permeability in

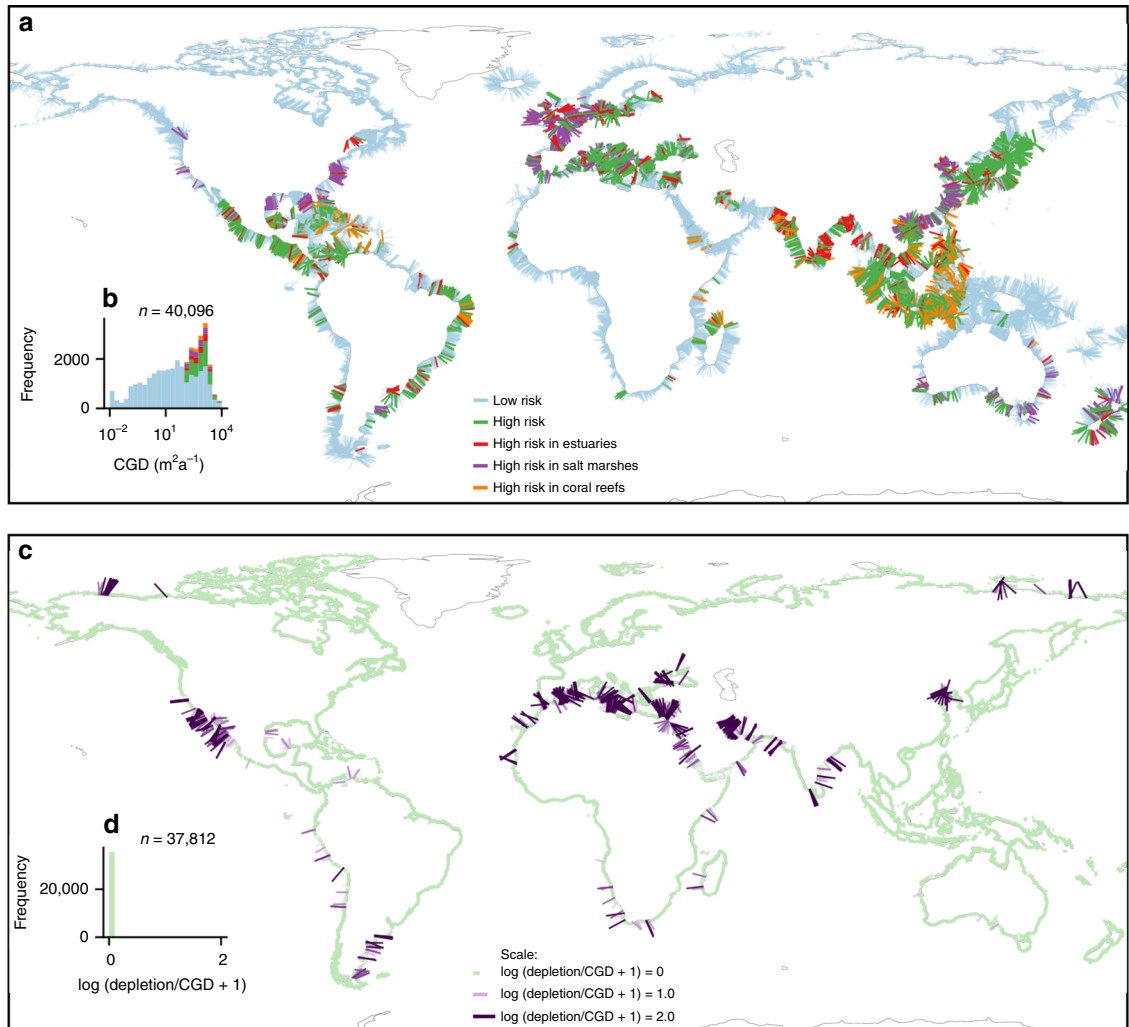

**Fig. 4 Maps of the eutrophication risk by coastal groundwater discharge and a comparison between coastal groundwater discharge and groundwater depletion.** Coastal groundwater discharge (CGD) can locally pose a eutrophication and pollution risk to coastal ecosystems (**a**). Although the majority of coastal watersheds groundwater depletion is lower than CGD, groundwater depletion locally exceeds CGD in approximately 13% of the global coastline (**b**). The coastal groundwater discharge, eutrophication risk, and groundwater depletion data that are shown here are available as Supplementary Data 3.

governing groundwater flow and discharge in coastal groundwater systems (Figs. 1, 2), which is also supported by previous work on the controls on groundwater flow in coastal aquifers[17].

For the Gulf and Atlantic coasts of the USA our model estimates a coastal groundwater discharge of 6.6 (0.1–11.3) km$^3$ a$^{-1}$, which is at the lower end of recent estimates by Zhou et al.[46] that range from 9.7 (7.2–12.0) to 27.1 (22.8–30.5) km$^3$ a$^{-1}$. These estimates are based on a series of regional groundwater models[47], include more detailed hydrogeology and permeability structure and may therefore be more accurate than our model estimates, especially at local scales. On the other hand, these models did not include solute transport and density-dependent flow and used a relatively coarse spatial discretization of 250 m. As a result, the partitioning of groundwater discharge around the coastline may not have been well resolved, and not including the fresh-salt water interface may have led to overestimation of coastal groundwater discharge.

## Conclusions

The assessment of coastal groundwater discharge reported here provides a high-resolution estimate of the distribution of this flux at a global scale that is consistent with the physics of density-driven groundwater flow in coastal groundwater systems. Model sensitivity analysis shows that for most coastal groundwater

systems the bottleneck for coastal groundwater discharge is their flow capacity, which is a function of permeability, the thickness of permeable units and topographic gradient, instead of the volume of water that is recharged in these systems. Our analysis shows that coastal discharge is subdivided in fresh submarine groundwater discharge and a roughly equally important component of terrestrial near-shore discharge, which has been overlooked in most previous analyses, and may have instead been lumped with fresh SGD in water budget analyses and model studies. The global flux is dominated by a small number of coastal watersheds that are distributed around the globe, including numerous locations at which coastal groundwater discharge has so far not been studied. In contrast to river discharge, coastal groundwater discharge is frequently unmonitored. However, our global analysis shows that locally coastal groundwater discharge can in many cases pose an equally high risk for coastal water quality and coastal ecosystems. In addition, groundwater discharge is in most cases relatively diffuse compared to surface water discharge and may therefore affect larger areas. Coastal groundwater discharge links terrestrial groundwater systems with coastal ecosystems, which means that changes in groundwater pumping or land use affect the flow of nutrients to coastal ecosystems[48,49], and should be taken into account in coastal environmental management. The estimates

provided here can help guide future research and monitoring of this water flux and its effect on coastal ecosystems. This is especially important as population pressures and increase in agricultural activity are likely to increase nutrient and contaminant inputs to coastal groundwater in many areas in the future. Furthermore, because groundwater flow rates are typically very slow, measures to improve groundwater quality onshore may take decades before they affect offshore water quality[50]. Therefore, quantitative estimates of coastal groundwater discharge are of key importance for identifying present and future risks to coastal water quality.

## Methods

**Modeling coastal groundwater discharge.** We simulated submarine and terrestrial groundwater discharge in coastal groundwater systems using a numerical model of coupled density-driven groundwater flow and solute transport. The model code, GroMPy-couple[51], is a Python shell around the finite element code escript[52,53] which has been used previously to simulate subsurface fluid flow[52]. We implemented an iterative scheme[54] to solve the fluid flow and solute transport equations and the equations of state for fluid density and viscosity. GroMPy-couple uses an implementation of a seepage boundary condition based on an existing implementation in the model code MODFLOW[14], which ensures a realistic and numerically stable partitioning between onshore and offshore groundwater discharge in models of coastal groundwater systems[14]. The model code simulates the flow of fresh (meteoric) groundwater, the mixing and recirculation of seawater at the fresh-salt water interface at the coast due to dispersion and the onshore and offshore discharge of groundwater. We did not model transient processes like tidal forcing of groundwater flow and wave setup, that are responsible for the bulk of the recirculation of seawater in coastal aquifers[55,56]. Note that the submarine discharge of fresh groundwater is relatively insensitive to transient flow induced by wave set-up and tides[56]. The model code has been validated by comparison with a salt water intrusion experiment[57] (Supplementary Fig. 9), analytical solutions of groundwater discharge[13,14] (Supplementary Figs. 10, 11), and model experiments using the widely-used model code SUTRA[58] (Supplementary Fig. 12). See Supplementary Note 1 for more details on the model approach and for more information on the validation of the model code.

**Model geometry.** Groundwater flow was simulated in a two-dimensional cross section of the subsurface. While we acknowledge that coastal groundwater flow is a three dimensional process, the computational demands of running large numbers of three dimensional models would be prohibitive, given the high spatial resolution required to accurately model density-driven flow and the constraints on timestep size imposed by numerical stability of modeling advective solute transport[59].

We assigned a constant linear slope to the terrestrial and marine parts of the model domain. The linear slope is a simplification. Testing more complex topographies, with for instance a lower slope of the near-shore parts that is often found in sedimentary settings, or conversely high relief and cliffs in erosional settings would increase the number of model runs that are needed to cover parameter space significantly, and would make the computational costs prohibitive. We therefore aimed to cover the first order effects of a linear topography.

The first 1000 m of the model domain are covered by seawater. The length of the landward size of the model was constrained by the watershed length described elsewhere in the Methods. The thickness of the model domain was kept at 100 m for the exploration of the parameter space, and was varied between 100 and 500 m for sensitivity analysis.

We applied a spatial discretization that varied from 3 m in a zone that extends 500 m offshore and 250 m onshore to 10 m on the landward boundary of the model domain. The zone with fine discretization is centered around the fresh-salt water interface that was calculated using an analytical solution[60]:

$$y^2 = 2\frac{\rho_f Q}{(\rho_s - \rho_f)K}x + \left(\frac{\rho_f Q}{(\rho_s - \rho_f)K}\right)^2, \qquad (1)$$

where $y$ is the depth of the interface below sea level (m), $K$ is hydraulic conductivity (m s$^{-1}$), which is calculated as $K = \rho_f g \kappa / \mu$, $\rho_f$ and $\rho_s$ are the density of freshwater and seawater, respectively (kg m$^{-3}$), $Q$ is the discharge rate of fresh groundwater (m$^2$ s$^{-1}$) and $x$ is distance to the coastline (m). The discharge term ($Q$) in Eq. (1) was calculated using a depth-integrated version of Darcy's law:

$$Q = Kb\frac{\partial h}{\partial x}, \qquad (2)$$

where $b$ is aquifer thickness (m), and $h$ is hydraulic head (m). In case the calculated discharge exceeded the total recharge volume (i.e., the product of the recharge and length of the model domain) into the system, discharge term $Q$ was capped to equal the recharge volume.

**Initial and boundary conditions.** A specified recharge flux boundary and a seepage boundary condition were applied to the upper model boundary at the terrestrial side of the model domain. For the seaward side of the model domain we applied a specified pressure that equals the load of the overlying seawater. Initial salinity was equal to seawater values of 0.035 kg kg$^{-1}$ under the seabed and in a saltwater toe that extends inland following Eq. (1). No flow was allowed over the left-hand and right-hand side of the model domain. Initial pressures were calculated by solving the steady-state version of the groundwater flow equation (Supplementary Note 1 and Eq. (1)).

The exchange of groundwater and surface water or evapotranspiration was simulated using a seepage boundary algorithm[14]. The seepage algorithm was chosen because it represents a more realistic upper boundary than often used fixed specified pressure or flux boundaries, while avoiding the computational cost of explicitly modeling evapotranspiration and surface-groundwater exchange. The seepage boundary condition was implemented using an iterative procedure. First, initial pressures were calculated by solving the steady-steady-state version of the groundwater flow equation, i.e., the groundwater flow equation with the derivative of pressure and concentration over time set to zero. For the first step a specified flux was assigned to the entire top boundary, which represents groundwater recharge from precipitation. Following the first iteration step, a specified pressure boundary was adopted at any surface node where the fluid pressure (P) exceeds 0 Pa. Fluid pressures were then recalculated again by solving the groundwater flow equation using this new boundary condition. Following each iteration step, the flux to the boundary nodes was calculated by solving the steady-state groundwater flow equation. Any surface nodes where the fluid pressure exceeds 0 Pa are added to the seepage boundary and are assigned a specified pressure of 0 Pa. Seepage boundary nodes that instead of outflow generate inflow into the model domain at a rate that exceeds the recharge rate were removed from the seepage boundary. To avoid oscillations in the solution, only the nodes that generate 10% or more inflow compared to the seepage node with the highest rate of inflow are removed from the boundary condition after each iteration. This iterative procedure is continued until the number of seepage nodes reaches a steady value.

The iteratively calibrated steady-state seepage boundary is used as an initial seepage boundary during the transient model runs. At each timestep, the active seepage boundary is inherited from the previous time step. The seepage boundary condition is removed for any node that has become a net source of water into the model domain. Any non-seepage node at the surface where the fluid pressure exceeds 0 Pa is added to the seepage boundary. This implementation of the seepage boundary condition ensures that the hydraulic head never exceeds the surface elevation, and that there is not more inflow than the specified recharge rate at any node at the surface. Both possibilities would be unrealistic, but allowed when either a specified flux or a specified pressure boundary condition would be used for the upper boundary. In addition, the seepage boundary method as implemented here avoids the use of an unknown and uncertain drain conductance parameter that is used in drain boundary conditions, which aim to provide a similar realistic upper boundary as the seepage boundary[14] but use a different algorithm to achieve this.

**Assumption of constant thickness and saturation.** The model domain was assumed to be fully saturated and the saturated thickness was constant in the model domain and independent of the modeled pressures and hydraulic head. This is a simplification that avoids the numerical instability and high computational costs of modeling unsaturated groundwater flow in combination with density-driven flow. At the same time, it makes comparisons between the individual model runs easier, because saturated thickness and transmissivity remain constant unless permeability is changed. In addition, when not adopted, the modeled thickness of the model domains for the different model scenarios would have to be sufficiently high to accommodate all possible modeled hydraulic gradients, which vary from values close to the highest modeled topographic gradients to values of near zero for the different model scenarios. This would again result in a prohibitively high number of model runs that would be required to cover the range found in coastal groundwater systems.

The assumption of a fully saturated subsurface does introduce errors in the modeled flow field. For models with a high topographic gradient there is a significant vertical flow component in our model setup, regardless of the shape of the watertable and the hydraulic gradient. However, for cases where permeability is high, the hydraulic gradient would relatively low and groundwater flow would in reality be nearly horizontal. The error in the partitioning between horizontal and vertical components of the flow vectors is expected to be equal in magnitude to the difference in topographic gradient and the modeled hydraulic gradient. The median difference between topographic and watertable gradient is 0.17%, and exceeds 5% in 2381 of the 40,082 modeled watersheds. This error however reduces to near zero close to the shoreline, where in all model experiments the watertable was located at or very close to the surface, and where therefore the assumption of fully saturated conditions is correct. Given the fact that the near-shore part of the model domain is by far the most critical for our model results on submarine and near-shore groundwater discharge, we expect the assumption of fully saturated conditions to not significantly influence the results reported here.

We used a constant thickness over the model domain. The thickness was varied between 50 and 500 m in a first sensitivity analysis. For the final model runs we adopted a standard thickness of 100 m, which is equal to the median aquifer

thickness that were used to compile data for a global permeability map[22], and is roughly equal to the thickness where the majority of young groundwater and active groundwater circulates following global compilations of radiogenic isotope data of groundwater[15,16]. Adopting a standard thickness of 100 m ensures that the modeled values of transmissivity (the product of permeability and thickness) are consistent with the global permeability map.

**Model runtime**. The transient models were run until a steady state was reached. We assumed that the model has reached steady state when the change in pressure is less than 1 Pa a$^{-1}$ and the change in solute concentration that is less than $1 \times 10^{-4}$ kg kg$^{-1}$ a$^{-1}$. The initial timestep size was 5 days and was increased by a factor of 1.03 after completing each timestep. To avoid numerical instability in solving advective solute transport, the maximum size of the timestep was adjusted automatically to not exceed the Courant-Friedrichs-Lewy (CFL) condition[61]:

$$CFL = q\Delta t/\Delta x, \qquad (3)$$

where $q$ is fluid flux (m s$^{-1}$), $\Delta t$ is timestep size (s) and $\Delta x$ is the size of one element (m) as calculated by escript[53]. We used a relatively low limiting value of CFL = 1.0 to ensure numerical stability for the large set of different models that we tested.

**Model sensitivity analysis and exploration of parameter space**. First, we performed a sensitivity analysis of submarine groundwater discharge by varying watershed length, topographic gradient, groundwater recharge and permeability in a range covers the values found in the geospatial analysis (Supplementary Fig. 2; Supplementary Table 3). The sensitivity analysis consisted of a total of 130 model runs. The parameters ranges are shown in Supplementary Table 4. The base case model used the median watershed length and groundwater recharge from the geospatial analysis, and a higher permeability ($10^{-12}$ m$^2$) and topographic gradient (2.5%) than the median watershed to better show the sensitivity of coastal groundwater discharge to the various parameters. In addition to the results shown in Fig. 1 we tested the effect of a realistic range of values of aquifer thickness, permeability anisotropy, and longitudinal dispersivity, which are variables for which no geospatial data was available. These additional results are shown in Supplementary Fig. 13 and discussed in Supplementary Note 2 and confirm that although these parameters are important, permeability and topographic gradient are by far the most sensitive parameters for coastal groundwater discharge.

Second, we conducted a number of model experiments ($n = 495$) to explore the parameter space for recharge volume (recharge multiplied by contributing area), topographic gradient and permeability. The ranges of these parameters are shown in Supplementary Table 5. The length of the model domain was kept constant at 3 km in all these model runs to reduce the number of runs require to cover parameter space. This does not affect model results because the model sensitivity analysis shows that changes in recharge volume by either changing the contributing area (watershed length) or the groundwater recharge rate have the same effect on modeled groundwater flow and discharge (see Fig. 1d, f). Apart from the recharge volume, permeability, and topographic gradient all other parameters were constant and followed the base case values listed in Supplementary Table 5. A number of runs ($n = 144$) did not converge to steady-state after a total number of 10,000 timesteps or were numerically unstable and were discarded. These consisted predominantly of models with a very low topographic gradient (<$10^{-3}$ m m$^{-1}$) or recharge rate (<0.01 m a$^{-1}$), where flow rates were so low that numerical precision affected the results.

Longitudinal dispersivity was kept constant at a value of 50 m and transverse dispersivity was assumed to be 0.1 times longitudinal dispersivity. Compilations of dispersivity data suggests that for the scales of the numerical models presented here longitudinal dispersivity varies between approximately 10 and 100 m, while transverse dispersivity is an order of magnitude lower[62]. However, some case studies in coastal aquifers have reported much lower numbers[63]. Nonetheless we have opted to use relatively high values of longitudinal and transverse dispersivity of 50 and 5 m, respectively. The reason is that lower values would strongly increase the computational costs, since for lower values of dispersivity one would have to decrease the grid cell size and increase the number of grid cells. Sensitivity analysis confirm that at least for values of longitudinal dispersivity of 50 m or more coastal groundwater discharge is relatively insensitive to dispersivity (Supplementary Fig. 13).

Permeability anisotropy (the ratio of horizontal over vertical permeability) was kept constant at a value of 10. For fractured crystalline rocks vertical permeability may exceed horizontal permeability, whereas for layered sediment sequences anisotropy can reach a factor of 100 or more[64]. We used a constant anisotropy value of 10 to strike a balance between these two end-members. Note that a more accurate implementation of permeability anisotropy in our models would require information on the orientations of fractures and bedding in coastal aquifers, which are currently not available at a global scale.

**Geospatial data analysis**. The model input was based on a geospatial data analysis of the controlling parameters of coastal groundwater discharge. We analyzed

watershed geometry[65], topographic gradients[66], permeability[22], and groundwater recharge[23,24] for 40,082 coastal watersheds globally (see Supplementary Fig. 2).

**Watershed geometry**. We used the geometry of coastal watersheds as a first order estimate of the size coastal groundwater systems. Note that in our model setup the area contributing to coastal discharge is determined by the model itself and the seepage boundary condition, and depends on the flow capacity of the subsurface and the recharge volume. In areas with high permeability, high relief, or low recharge the watertable can be decoupled from topography and regional flow that bypasses the nearest discharge points can be significant[67,68], which means the contributing area could be larger than the size of the model domain. A published comparison of recharge and discharge estimates in a large number of river basins indicates that for the majority of basins the regional flow component is less than 50%[69]. To cover the uncertainty of the size of the contributing area in our calculations of coastal groundwater discharge, we used a value of two times the size of surface watersheds as a maximum estimate.

First, 40,082 coastal watersheds were selected using global watershed[65] and coastline[70] datasets. Second, the local watershed divide to the ocean for each watershed was identified using GIS tools. The local watershed divide was defined as the boundary between each watershed and adjacent non-coastal watersheds. The representative length scale was taken as the mean distance of the water divide to the coastline. For watersheds that were only bound by other coastal watersheds we used the mean distance of the centroid of the watershed to the coastline as a representative length scale. Note that the global analysis may underestimate coastal groundwater discharge in several tropical islands in the Pacific, since islands such as Hawaii and Mauritius are missing from the global watershed database[65] that supports the analysis.

**Permeability**. Permeability of each coastal watershed was extracted from a global dataset of near-surface permeability (up to ~100 m depth)[22]. The permeability map is based on a high-resolution global map of surface lithology[71] and a compilation of large-scale permeability estimates of near-surface geological units[22]. We made two changes compared to the permeability map with the aim of ensuring that the modeled coastal groundwater discharge is a high, but still conservative and realistic estimate.

In the global permeability map, areas in which the lithology consisted of mixed unconsolidated sediment or unconsolidated sediments with an unknown grain size were assigned a relatively low permeability ($10^{-13}$ m$^2$)[22]. This value is below the threshold for generating significant coastal groundwater discharge, except for settings with a very high topographic gradient. However the multimodal distribution of this unit[22] suggests that in many cases this unit contains coarse grained sediments. In layered unconsolidated sediments, the effective permeability is likely to be close to the value of the most permeable sub-unit, whereas the permeability value assigned in the global permeability map is the mean permeability on a log scale. We instead assume that the permeability for coarse-grained unconsolidated sediments ($10^{-10.9}$ m$^2$) is more appropriate to simulate regional groundwater flow in coastal aquifers that were classified as mixed or unknown unconsolidated sediments in the global permeability map.

The permeability of carbonates in the global permeability map is likely underestimated in coastal areas with strong karstification. Coastal carbonates are predominantly karstified[72], in part because the fresh-salt water mixing zone at the coastline promotes dissolution and the formation of permeable karst conduits[73]. We therefore adopted a higher permeability estimate equal to the value reported in the global permeability map plus one standard deviation ($10^{-10.3}$ m$^2$). While we acknowledge that this value is highly uncertain as a global average for coastal karst aquifers, and that in reality their permeability is likely to be highly variable, this value is in line with reported values of regional-scale permeability in coastal karst aquifers[74–76].

**Topography**. Elevation data[66] was extracted for each coastal watershed. We calculated distance to the coastline and elevation for each point in the elevation raster. We then calculated the average and standard deviation of the topographic gradient by dividing elevation by the distance to the coast. In addition, we calculated the topographic gradient for raster cells that contained a stream. The locations of streams were obtained by grouping raster cells for which the distance to the coast is the same within a range that is equal to the size of one raster cell, and then selected the raster cell with the lowest elevation for each distance value.

**Additional geospatial datasets**. We extracted a number of global datasets for a comparison with the calculated submarine and terrestrial discharge, including river discharge[23], potential evapotranspiration[27], the elevation of tide and storm surges[77], and watertable gradient, which was quantified using a global model of watertable depth[78] and global elevation datasets[66].

**Representative topographic gradient**. The numerical models used for the model sensitivity analysis and the global estimate of coastal discharge use a linear topographic gradient. The topographic gradient is important because it sets a maximum for the hydraulic gradient in each watershed and because it governs the partitioning of coastal groundwater discharge between NGD and fresh SGD (Fig. 1e). We

explored which metric can be used as a representative linear topographic gradient by comparing coastal discharge for a series of model runs that include into account the real topography with cross-sectional groundwater models that only include a linear topography. An example comparison of these two model approaches for a watershed is shown in Supplementary Fig. 14. The comparison shows that for most of the tested watersheds coastal groundwater discharge can be represented by cross-sectional models that use the average topographic gradient of an entire watershed, or the average topographic gradient of stream channels in each watershed (Supplementary Fig. 15). See Supplementary Note 3 for more details on these model experiments. Based on these results we used the average topographic gradients in coastal watersheds as a best estimate in further model experiments and the average topographic gradient of streams as a lower bound for the uncertainty range of coastal groundwater discharge.

**Quantification of global coastal groundwater discharge flux.** Global fluxes of CGD, fresh SGD, and NGD were obtained by linear interpolation of 351 model results to the 40,082 coastal watersheds (see Fig. 2 and Supplementary Figs. 3, 4). The interpolation is based on a comparison of permeability, topographic gradient, and recharge volume, which is the product of groundwater recharge and the size of the contributing area, for the model results and the geospatial data of coastal watersheds. For each watershed CGD was calculated by linear interpolation of the modeled CGD values of the model runs with the closest values of recharge volume, permeability and topographic gradient. A comparison of the modeled and interpolated values of CGD is show in Fig. 2. For $n = 13,025$ watersheds the parameter values were located outside the bounds of the parameter combinations tested by the model runs. These were predominantly watersheds with in which permeability was lower than the lowest modeled permeability value ($<10^{-16}$ m$^2$). In these cases we used a nearest neighbor algorithm[79] to assign the CGD value of the closest model run to the watershed. The average difference between linear interpolated and the nearest modeled CGD value in log units was 0.26. We repeated the same interpolation procedure to calculate fresh SGD (Supplementary Fig. 3), NGD (Supplementary Fig. 4), maximum discharge flux and the horizontal and vertical extent of the submarine and terrestrial discharge zone for each coastal watershed.

We used reported ±1 standard deviation uncertainty of permeability[22], the differences between two alternative recharge datasets[23,24] and the differences in the topographic gradients between elevation grid nodes covered by streams and the entire coastal watershed to calculate minimum and maximum estimates of the discharge fluxes. The interpolation yielded two dimensional cross-sectional fluxes in units of m$^2$ a$^{-1}$. Volumetric fluxes were calculated by multiplying the flux by the length of the coastline for each watershed. While the coastline length is in principle a fractal property with a value that depends on the scale of observation, in our case a value was chosen such that the product of the coastline length and the representative length scale of each watershed equaled the area of each watershed. The interpolated values of groundwater discharge were compared to published values of surface runoff[23], potential evapotranspiration[27], groundwater depletion[45], and tide and storm surge elevation[77], which were assigned to each watershed using GIS tools. For the comparison of the spatial distribution of CGD and runoff, evapotranspiration or depletion, all fluxes were summed up at the watershed scale.

**Model-data comparison.** Comparison between modeled and measured average hydraulic gradients in 336 coastal watersheds with water level observations from a global dataset[78] shows that the ratio of the modeled values over the measured values is 1.08 (see Supplementary Note 4 and Supplementary Fig. 6) when using the best estimates of permeability, recharge, and topographic gradient. This shows that the model provides a realistic estimate of coastal groundwater flow. Overall ~60% of the local variability in water table gradient is captured. The remaining variation is likely due to the large spatial scale of our models (on average 11 km) and the limitations of the global datasets supporting our analysis. A qualitative comparison with locations of reported ecosystem impacts by fresh SGD and reported use of fresh SGD shows that in all of these locations the modeled fresh SGD is much higher than the median of all coastal watersheds globally (see Supplementary Tables 1, 2). Comparison to a selection of ten published local estimates of fresh SGD shows that for five studies the modeled and reported fresh SGD values fit within uncertainty bounds (Supplementary Fig. 16a; Supplementary Table 6). However, for the remaining locations the reported values strongly exceed the modeled values, but the reported fresh SGD also exceeds the total groundwater recharge in adjacent coastal watersheds (Supplementary Fig. 16b). This in spite of the fact that all of the reported values are located in watersheds with perennial streams that also discharge a large part of the overall groundwater recharge. This suggests that these studies strongly overestimate fresh SGD. The frequent inconsistency of fresh SGD estimates with onshore groundwater budgets has been noted earlier by several authors[11,80] and may be due to uncertainties in methods to quantify fresh water discharge[81,82] or biased selection and reporting of study sites for SGD[1].

**Quantification of solute transport.** First-order estimates of the global solute flux that is transported by fresh SGD were calculated by multiplying the calculated global fresh SGD flux with previously reported average values for the concentrations of nitrogen[28,29], carbon[83], silica[7], and strontium[8] in coastal groundwater. These

estimates were compared to published values of the solute flux by surface runoff and earlier estimates of the contribution of fresh SGD[7,8,30,83]. These estimates are based on published compilations of data predominantly from the US, which we consider as first order estimates for concentrations in groundwater at a global scale that are relatively uncertain. The estimate of carbon was based on the global average DIC in soil water of 15 mg L$^{-1}$ as calculated from reported global DIC fluxes to the groundwater table and groundwater recharge by Kessler and Harvey[84]. The average strontium concentration in groundwater of 2.9 μM reported by Beck et al.[8] is based on compilations of groundwater strontium data and an extrapolation based on a global lithology map. The nitrogen concentration of 2 mg L$^{-1}$ was based on a reported median value for a large dataset from the US[39]. The silica concentration was equal to a value reported by Frings et al.[85] based on a compilation of groundwater data predominantly from the USA. The calculated values of the global solute flux by SGD were compared to published values of the solute flux by surface runoff and earlier estimates of the contribution of fresh SGD[7,8,30,83]. Note that our estimates assume conservative transport and do not take into account fluid-rock interaction and microbial activity in coastal aquifers. Several studies have shown that intensive reactions happen in the mixing zone between fresh-and saltwater (subterranean estuary). Despite the effects that were described for individual local systems, no large scale estimates of the effect of subterranean estuaries are known to the authors, therefore this effect cannot be quantified in this study.

**Estimation of eutrophication risk.** The risk of eutrophication of coastal ecosystems was estimated by comparing locations of high groundwater discharge with high agricultural nitrogen inputs in coastal watersheds. Areas with high risk were identified if CGD exceeded a threshold of 100 m$^2$ a$^{-1}$ and nitrogen inputs[86] exceed 10 kg ha$^{-1}$. The threshold value for CGD is based on a review of locations with reported impacts of coastal groundwater discharge on ecosystems (see Supplementary Note 4 and Supplementary Table 7). The threshold value for nitrogen input corresponds to values that have historically led to strongly elevated nitrogen concentrations in groundwater in Europe and North America[39,40] that have contributed to the eutrophication of terrestrial and nearshore ecosystems[41]. Given the very high sensitivity of coastal ecosystems and especially coral reefs to nitrogen[87], this threshold value is likely to result in a relatively conservative estimate of eutrophication risk. The calculated locations of sites with high eutrophication risk was compared with the location of sensitive ecosystems such as estuaries[42], salt marshes[43], and coral reefs[44]. We acknowledge that eutrophication of marine ecosystems is a complex function of nutrient input, transport, denitrification, and mixing with seawater. The eutrophication risk reported here should be considered as a first order estimate that can guide follow-up studies. Finally we also compared the modeled coastal groundwater discharge fluxes to groundwater depletion by using published groundwater depletion rates[45] and multiplying these rates with the area of coastal watersheds.

## Data availability
The results of the model sensitivity analysis shown in Fig. 1 is available as Supplementary Data 1. The results of the model parameter space exploration shown in Fig. 2 is available as Supplementary Data 2. The results of the geospatial analysis shown in Figs. 1, 2, and the interpolated coastal groundwater discharge fluxes for the global watersheds shown in Figs. 3, 4 are available in shapefile format as Supplementary Data 3. The data files are also available on Pangea[88].

## Code availability
The model code used to simulate coastal groundwater discharge (GroMPy-couple) is available at GitHub (https://github.com/ElcoLuijendijk/GroMPy-couple), along with the parameter files required to replicate the results of this study. The source code has also been published at Zenodo[51].

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

## Acknowledgements

E.L. and T.G. were supported by NSERC and CIFAR. N.M. was supported by the BMBF project SGD-NUT (Grant #01LN1307A). We acknowledge support by the Open Access Publication Funds of the University of Göttingen. We would like to thank Lou A. Derry for a very helpful review of an early version of this manuscript.

## Author contributions

TG conceived this study. E.L. and T.G. designed the methods. E.L. constructed the model code and conducted the geospatial analysis and numerical modeling. E.L., T.G., and N.M. contributed to interpretation and discussion of the results and writing the manuscript.

## Competing interests

The authors declare no competing interests.
