## [Peer Review File · Nature Communications]

Reviewers' comments:

Reviewer #1 (Remarks to the Author):

This article tries to show global distribution of coastal groundwater discharge to the ocean, as well as nutrient loads by groundwater to the near-shore terrestrial coast and the coast under the sea level. The paper emphasizes the risk for pollution and eutrophication through groundwater discharge to the ocean in the estuaries, salt marshes and coral reefs, however the groundwater can carry dissolved material as nutrients for feeding flora and fauna in the coastal estuaries and ocean. The authors have to show both positive and negative impacts of terrestrial groundwater discharge to the ocean with evidences.

The followings are individual comments:

1) 42-44: Definition of 2) "near-shore terrestrial groundwater discharge (NGD) that is fresh groundwater discharging above the mean sea level in the first hundreds of meters near the coastline" makes readers confusions or misunderstanding. The tidal and sea level changes cause the spatial and temporal changes of 1) Fresh SGD and 2) NGD. Author need clear definition of the difference between 1) and 2), in terms of tidal and sea level changes.

2) 41-44: Inconsistence of the numbers of Fig 1 and the text.

3) 98-109: Uncertainty: Authors need to show here the statistical numbers of uncertainty of permeability, representative topographic gradient of coastal watersheds, groundwater recharge and the size the area that contributes to fresh SGD.

4) 119-121: The authors mention "which is much smaller than the up to 100% contribution suggested by some recent studies that extrapolated global inputs from local and regional-scale estimates.⁴⁻⁶" Authors should mention here why the results were different from literatures. Is this because scale issue or insufficient data or including recirculated water?

5) 134-138: Authors mention "Model experiments demonstrate that the partitioning of coastal groundwater discharge in submarine and terrestrial discharge is highly sensitive to the local topographic gradient (Fig. 1d), and therefore the total coastal groundwater discharge is a more robust estimate than the onshore and offshore components." If so, why do not you show in Fig. 3 for not only coastal groundwater discharge (CDG) with one case of topographic gradient, but also CDG with different topographic gradient scenario ?

6) 169-173: The authors mention "Eutrophication risk is defined by nitrogen application in coastal watersheds that exceeds 10 kg ha⁻¹ and coastal discharge that exceeds 100 m² a⁻¹. The threshold value for nitrogen input corresponds to values that have historically led to strongly elevated nitrogen concentrations in groundwater in Europe and North America^{31,32} that have contributed to the eutrophication of terrestrial and nearshore ecosystems³³." However, the literatures are limited to Europe and North America, so author should mention the limitation of the application for the other areas and other coastal environments. Low volume of groundwater discharge (less than 100m² a⁻¹) may also cause eutrophication in near-shore terrestrial and submarine coastal environment.

7) 189-193: Authors mention "A comparison of the calculated coastal groundwater flux and published values of groundwater depletion in coastal watersheds³⁸ shows that in most coastal groundwater systems are not associated with depletion (Fig. 4b). Groundwater depletion is concentrated in semi-arid regions and in 13% (8% - 19%) of the global coastal watersheds groundwater depletion exceeds coastal groundwater discharge." This comparison is good for future management of groundwater in particular semi-arid area. Reviewer recommends authors to show

not only ratio of exceeded coastal line, but also exceeded global volume of CGD and groundwater depletion.

8) 226-227: The authors mention "groundwater discharge is relatively diffuse compared to surface water discharge and may therefore affect larger areas." However, is there any evidence to support this sentence ? Authors should show the area which is affected by river is smaller than that by CGD.

Reviewer #2 (Remarks to the Author):

In their manuscript, Luijendijk et al. present a global analysis of fresh submarine groundwater discharge (SGD), onshore discharge to the coast (NGD), and saline SGD. They include an analysis relating these groundwater discharges to river discharge, and they consider the potential impact on nutrient flow to coasts. This is an important analysis based on a numerical sensitivity study that involves two-dimensional density-dependent flow and salt transport near the coast. I have two major concerns with this study.

First, the study includes many novel aspects that could each warrant full article-length discussion: 1) a global analysis of fresh SGD, NGD, and saline SGD (the main text presented here), 2) a new sensitivity study that describes why fresh SGD, NGD, and saline SGD should be high or low in different coastal systems (e.g. Figure 1, S10)—the separation of these three fluxes is itself a relatively novel idea, 3) validation of a new code for variable density flow (e.g. Figures S2 and S3), 4) assessment of potential solute fluxes to the coast (e.g. Figure 4 and Table 1). While the first message is of high-impact (and the last on nutrient inputs), it cannot easily be understood without a mechanistic, process-based explanation of the sensitivity results (#2), which are currently scattered throughout sentences in the main text and supplementary text. Most of the text in L 53-81 is difficult to follow because it depends on the behavior of the sensitivity study, which is never fully described in a mechanistic, process-based way, even in supplementary text. The physics of this problem would be better presented in a full-length journal article that allots ample space for conceptual development, numerical methods, and behavior of the results.

Second, the authors suggest that large-scale spatially distributed estimates of fresh SGD have only been published for the US. Another recent study reports estimates for the globe:

Zhou, Y. Q., Sawyer, A. H., David, C. H., & Famiglietti, J. S. (2019). Fresh submarine groundwater discharge to the near-global coast. *Geophysical Research Letters*, 46.

<https://doi.org/10.1029/2019GL082749>

Many of the results appear to be similar at first glance (for example, concentration of fresh SGD into a small portion of the total coastline, which the authors term "hotspots", which are often located in tectonically active areas). This study is independent and clearly doesn't duplicate the Zhou study because Luijendijk et al. use different methods, break groundwater discharge into components of fresh SGD, NGD, and saline SGD, and consider potential nutrient fluxes. However, the results can and should be directly related.

Minor suggestions are below:

- L 39: See Zhou et al. (2019)

- L 47, 49, and throughout – it took me a while to understand whether "onshore" groundwater discharge and "coastal" groundwater discharge were the same and equivalent to fresh SGD plus NGD (#1 and #2 in L 41-44) or different, and I'm still not sure I have assumed the correct relationships. The authors need to use consistent terminology throughout the main text or define equivalent terms explicitly.

- L 63 and other lines in this paragraph lack context for understanding without a presentation of the sensitivity behavior first

- L 75: I would disagree that NGD has been overlooked, as it is probably included with fresh SGD in any approach based on a water budget (and is essentially the focus of work by Destouni et al., 2008 on ungauged coastal discharge).
 - L 89: this observation ties naturally to the results of Michael et al. (2013)—consider citing
 - L 110 and following paragraph—I was unclear on whether NGD is included. Most sentences refer to fresh SGD, but there are also sentences that refer to “terrestrial contributions” that should consider NGD. Even if much NGD is evapotranspired, solutes remain on the landscape, are transported during high tide and flood events to wetlands and estuaries, and exchange with the ocean over different timescales.
 - L 147: How are groundwater and river fluxes related (over what areas), as rivers are technically point sources to the coast? Is there an integration window for summing up the river fluxes, and if so, what is that coastline length?
 - L 169: These results would benefit from comparison with Beusen, which is cited in the Supplementary Text.
 - L 212-213: see Zhou et al. (2019)
- In the supplemental material:
- L 151: How deep is most groundwater extraction near the coast? A discussion of how different aquifers are treated in the homogeneous models and where extraction tends to come from might help here.
 - L 169: I could not easily find the description of the landward boundary condition but assume it is specified head.
 - L 303: The behavior of the sensitivity analysis needs to be presented mechanistically (how does each component of discharge behave under changes in recharge, permeability, etc, and why) in order to understand the main text.
 - L 424: The implications of assuming no rock-water interactions or reactions near the sediment-water interface should be discussed using the marine geochemistry literature.
 - L 491-494: Most seepage meter studies are conducted in clearly “offshore zones” and capture fresh SGD because otherwise the meters would be exposed during high tide (they are usually deployed for days in advance of measurement to establish steady flow conditions).

References

- Destouni, G., F. Hannerz, C. Prieto, J. Jarsjö, and Y. Shibuo (2008), Small unmonitored near-coastal catchment areas yielding large mass loading to the sea, *Global Biogeochem. Cycles*, 22, GB4003, doi:10.1029/2008GB003287.
- Michael, H. A., C. J. Russoniello, and L. A. Byron (2013), Global assessment of vulnerability to sea-level rise in topography-limited and recharge-limited coastal groundwater systems, *Water Resour. Res.*, 49, 2228–2240, doi:10.1002/wrcr.20213.

Reviewer #3 (Remarks to the Author):

Review

The flow of fresh groundwater and solutes to the world’s oceans and coastal ecosystems
Luijendijk et al.

This manuscript covers an interesting topic to the readers of Nature Communications. The authors show that coastal groundwater discharge can be a significant contributor to the water and chemical budgets of coastal catchments. The simulated flux is highly variable, with hotspots in areas with higher permeabilities, and could result in a high eutrophication risk. This study is the first estimate at the global scale of coastal groundwater discharge and near-shore discharge, therefore includes also information of locations where observations or models are not available.

Although I think this paper is discussing excellent research, I do think methods and results should

be more clearly described and discussed to allow the reader to fully understand the importance and novelty of this research. In my comments I focus on a few aspects of the paper where I think a more detailed and/or more clear explanation is needed.

Point 1

The following definitions are introduced in the introduction:

- Submarine groundwater discharge is the flow of fresh or saline groundwater to the oceans.
- Fresh submarine groundwater discharge (fresh SGD) is the fresh component of submarine groundwater discharge .
- Coastal groundwater discharge is the total flux of groundwater to the ocean and can be divided into 1) fresh SGD, 2) near-shore terrestrial groundwater discharge (NGD), 2) recirculating sea water.

The three fluxes, summing up to coastal groundwater discharge are shown in Figure 1. These definitions are clear, also for nonexperts.

In line 47 is written “.. of onshore and offshore groundwater discharge ... coastal discharge”. Are these onshore and offshore fluxes the same as submarine groundwater discharge (fresh or saline) and near-shore terrestrial groundwater discharge? Is ‘coastal groundwater discharge’ meant instead of coastal discharge? (or if not, coastal discharge needs to be explained).

In line 51-53 is written “In addition to submarine groundwater discharge we quantify near-shore terrestrial discharge”. This is confusing as in the previous line it is stated that coastal groundwater discharge was quantified, including fresh SGD, and NGD already. Is this an addition compared to what is estimated in previous studies? Also, is near-shore terrestrial groundwater discharge meant here instead of terrestrial discharge? (if not, near shore terrestrial discharge needs to be explained as well).

Additionally, in Figure 2a different abbreviations are used for fresh SGD and NGD (namely FSGD and NTGD respectively). Abbreviations should be used consistently. Also, the numbering of fluxes in the text is not the same as in the figure (namely numbering in the figure is 3), 1), 2) if the same numbering as in the text is used). I suggest to make this consistent too. In Figure 2b ‘submarine discharge’ is used, this should be submarine groundwater discharge (as this is how it is called throughout the text).

Lastly, are the ‘density-dependent groundwater models’ the same as the series of ‘numerical-models of groundwater flow’? And is the ‘density-dependent model’ (L56) the same as the density-dependent groundwater models?

This are all examples from the introduction part of the paper. I strongly recommend to go through to text and correct these, and similar, inconsistencies in the definitions used.

Also, from this introduction it is not so clear (or convincing) yet what the big selling point is of this research. For example, in this introduction it is not mentioned that this study provides a global estimate. I would suggest to add one or two lines stressing what is new in this study to help the reader understand the added value of this research.

Point 2

Overall it is hard to follow the ‘story line’ of the paper; it is not always clear what was done and to which purpose and the discussion of the results is not always easy to follow.

For example, the section ‘controls on coastal groundwater discharge’ start with ‘sensitivity analysis...’, but what is tested in this sensitivity analysis and how, is not discussed (at least not in the main text). One or two lines briefly stating which parameters were tested, how many runs were done, which data was used etc, would help the reader a lot to interpret and value the results correctly.

Also, it is not entirely clear that the examples given in section ‘controls on coastal groundwater discharge’ are actually two examples of runs done as part of the sensitivity analysis (or did I understand this wrong?).

In addition to the examples, in lines 56-59 is stated that fresh SGD is insignificant, it might be more logic to first discuss the average scenario where indeed fresh SGD is insignificant and second the exception case where it is 29% of the total recharge flux. Minor comment: in the average example a permeability and gradient are given, in the 29% example only a permeability is given, is the gradient not important?

As a second example, I did not have any clue what was done to calculate coastal groundwater discharge by paring the results of 351 model runs to geospatial data of 40,082 coastal watersheds (192-95). From the methods it became clear that linear interpolation was used. I would suggest to rephrases the sentence in the main text to something like this: "... by linear interpolation of 357 model results to geospatial data of 40,082 coastal watershed, covering information of permeability, topographic gradient, and recharge input. Minor comment: instead of recharge input you could also use recharge volume, which is more descriptive for recharge input.

My general point here is that, although the details of the used methods are hidden in the method sections and Supplementary information, I at least want to understand the basics of each method and step in the research while I am reading the main text.

Another general point: I often have the feeling results of one analysis (e.g sensitivity, or interpolation) can be clustered more together, and then be discussed. For example, in the section 'Global coastal groundwater discharge' the results of the calculation of coastal groundwater discharge, fresh SGD and NGD, are discussed and many percentages and volumes are given scattered over this whole section. Why not structure it a bit more, for example something like this: global coastal groundwater discharge, fresh SGD and NGD, was calculated. Only 4% of recharge is discharged offshore. Of the total SGD flux, 0.06% is fresh SGD, which amounts to 0.2% of the river discharge to oceans. Xx% of recharge is discharged on-shore, which is 0.4% of the global river discharge. And Tthen discuss.

I recommend to go through the text and especially focus on clear writing: make it easy for the reader to follow the story line of this paper.

Minor comments

L30-34: The fresh component of submarine groundwater discharge is critical, due to its high solute and nutrient load³, and has been estimated to be up to 10% of the river discharge to the world's oceans¹ and to equal the inputs by rivers to the for solutes such as carbon⁴, iron⁵, silica^{6,7} and strontium⁸, and potentially buffers ocean acidification with groundwater alkalinity⁹.

In Line 37 "high discharges" is this high submarine discharges or stream discharges or ...

L36: "...locally, it is difficult to derive a global estimate from these point estimates,...": which point estimates; are all local information observations? Or also modelling studies (if this is true I am not sure to call it point estimates)

L50: "at the global scale"

L53: "coastal water budgets ... " : coastal water and chemical? budgets ...

L57: remove comma before and

line 64 "... which is lower ..."; this is a bit too vague, give the value or percentage you compare it to. And what do we learn from this; right now, this is presented as a result, but is should be part of a discussion.

L74-77: "...but has to our knowledge larger scales": stated on the previous (introduction) section already, the few new information points should be places in the introduction, can be repeated in the conclusion again.

L77-80: two-dimensional cross-sectional model are the same as the density dependent groundwater models (L47)?

L84: relative to what, the area of watersheds globally?

L84-91: repetition of the previous section, not needed.

L98 "The uncertainly/variability(?) of the estimated global fresh SGD flux".

L99: which data was used? (add reference)

L104: add references for the global models.

L191-193: is it groundwater depletion per year what is meant? And averaged over the globe?

L223: "...not been explored". Is this referring the missing observations or (also) missing modelling?

L226: "serious risk": high risk

Figure captions of Fig.3 and Fig.4 both miss a brief title that applies to both figure panels. Actually, for figure 3 this title is in the second part "Global maps ...": move this part up.

Reply to reviewer's comments for manuscript "The flow of fresh groundwater and solutes to the world's oceans and coastal ecosystems"

Author's comment: The reviewers comments are reproduced below. Detailed replies to each of the reviewer's comments can be found below and are marked in blue

Reviewer #1 (Remarks to the Author):

This article tries to show global distribution of coastal groundwater discharge to the ocean, as well as nutrient loads by groundwater to the near-shore terrestrial coast and the coast under the sea level. The paper emphasizes the risk for pollution and eutrophication through groundwater discharge to the ocean in the estuaries, salt marshes and coral reefs, however the groundwater can carry dissolved material as nutrients for feeding flora and fauna in the coastal estuaries and ocean. The authors have to show both positive and negative impacts of terrestrial groundwater discharge to the ocean with evidences.

The followings are individual comments:

1) 42-44: Definition of 2) "near-shore terrestrial groundwater discharge (NGD) that is fresh groundwater discharging above the mean sea level in the first hundreds of meters near the coastline" makes readers confusions or misunderstanding. The tidal and sea level changes cause the spatial and temporal changes of 1) Fresh SGD and 2) NGD. Author need clear definition of the difference between 1) and 2), in terms of tidal and sea level changes.

We have added a sentence to clarify the difference to lines 41-42: "Note that we use the long-term (annual) mean sea level to separate SGD and NGD, and not the high tide line as used by some previous studies."

2) 41-44: Inconsistence of the numbers of Fig 1 and the text.

We have changed the numbers in the figure to match the text.

3) 98-109: Uncertainty: Authors need to show here the statistical numbers of uncertainty of permeability, representative topographic gradient of coastal watersheds, groundwater recharge and the size the area that contributes to fresh SGD.

We have added numbers for the uncertainty of these parameters in lines 149 to 158

4) 119-121: The authors mention "which is much smaller than the up to 100% contribution suggested by some recent studies that extrapolated global inputs from local and regional-scale estimates.4-6" Authors should mention here why the results were different from literatures. Is this because scale issue or insufficient data or including recirculated water?

We have added a sentence to clarify why these previous studies have concluded such high solute fluxes to the oceans in line 183-185 : “. The difference is most likely the result of scaling up high local rates of SGD, and the difficulty of separating fresh and recirculated SGD in measurements “

5) 134-138: Authors mention “Model experiments demonstrate that the partitioning of coastal groundwater discharge in submarine and terrestrial discharge is highly sensitive to the local topographic gradient (Fig. 1d), and therefore the total coastal groundwater discharge is a more robust estimate than the onshore and offshore components.” If so, why do not you show in Fig. 3 for not only coastal groundwater discharge (CDG) with one case of topographic gradient, but also CDG with different topographic gradient scenario ?

The statement concerns the partitioning between terrestrial and submarine groundwater discharge, which is highly sensitive to the topographic gradient. However, the total coastal groundwater discharge (CGD) estimate is much less sensitive to topographic gradient. We discuss the uncertainty range of coastal groundwater discharge and the sources of uncertainty extensively in lines 149 to 159. As discussed here topographic gradient is only one of the sources of uncertainty of this parameter and is less important than permeability.

6) 169-173: The authors mention “Eutrophication risk is defined by nitrogen application in coastal watersheds that exceeds 10 kg ha⁻¹ and coastal discharge that exceeds 100 m² a⁻¹. The threshold value for nitrogen input corresponds to values that have historically led to strongly elevated nitrogen concentrations in groundwater in Europe and North America^{31,32} that have contributed to the eutrophication of terrestrial and nearshore ecosystems³³.” However, the literatures are limited to Europe and North America, so author should mention the limitation of the application for the other areas and other coastal environments. Low volume of groundwater discharge (less than 100m² a⁻¹) may also cause eutrophication in near-shore terrestrial and submarine coastal environment.

Agreed, we now discuss the limitations of having data from North America and Europe only in line 231-232: “Note that the lack of data outside of North America and Europe make the extrapolation relatively uncertain, and the estimate here should be considered a first order estimate “

7) 189-193: Authors mention “A comparison of the calculated coastal groundwater flux and published values of groundwater depletion in coastal watersheds³⁸ shows that in most coastal groundwater systems are not associated with depletion (Fig. 4b). Groundwater depletion is concentrated in semi-arid regions and in 13% (8% - 19%) of the global coastal watersheds groundwater depletion exceeds coastal groundwater discharge.” This comparison is good for future management of groundwater in particular semi-arid area. Reviewer recommends authors to show not only ratio of exceeded coastal line, but also exceeded global volume of CGD and groundwater depletion.

Agreed, we have added the total volume of depletion in coastal watersheds and the depletion in the coastal watersheds where depletion exceeds CGD to lines 251-255.

8) 226-227: The authors mention “groundwater discharge is relatively diffuse compared to surface water discharge and may therefore affect larger areas.” However, is there any evidence to support this sentence ? Authors should show the area which is affected by river is smaller than that by CGD.

Globally half of the coastlines are small deltas and tidal systems (Dürr et al., 2011), which both have mostly local influence of river effluent. Only 0.7% of the coast is associated with large rivers (contrary to 39% of the river basins that are associated with the large rivers) (Dürr et al., 2011). Another 35% of the coastline are fjords, arctic or karst (Dürr et al., 2011) - neither of which are strongly influenced by rivers. The coastal length impacted by river plumes depends mostly on the shape of the plume, which in itself is defined by density contrast, currents and wind. In some occasions, river plumes can stretch along the shore (Kourafalou et al., 1996;). So while often even large river plumes do not influence long stretches of coast, because they quickly move offshore, in these occasions, rivers would affect a longer coastline.

And while previous SGD studies were uncertain in the volumes of global fresh SGD, most agree that it is an ubiquitous feature of coastal areas (e.g. Rodellas et al., 2012), contrary to rivers. Therefore we think our statement is justified that the length of the coastline that is affected by SGD is longer than that affected by rivers.

References

Dürr, H.H., Laruelle, G.G., van Kempen, C.M., Slomp, C.P., Meybeck, M., Middelkoop, H., 2011. Worldwide Typology of Nearshore Coastal Systems: Defining the Estuarine Filter of River Inputs to the Oceans. *Estuaries and Coasts*, 34(3): 441-458.

Kourafalou, V. H., Oey, L.-Y., Wang, J. D., & Lee, T. N. (1996). The fate of river discharge on the continental shelf: 1. Modeling the river plume and the inner shelf coastal current. *Journal of Geophysical Research: Oceans*, 101(C2), 3415–3434. doi:10.1029/95jc03024

Rodellas, V., Garcia-Orellana, J., Masque, P., Feldman, M., Weinstein, Y., 2015. Submarine groundwater discharge as a major source of nutrients to the Mediterranean Sea. *Proceedings of the National Academy of Sciences of the United States of America*, 112(13): 3926-3930.

Reviewer #2 (Remarks to the Author):

In their manuscript, Luijendijk et al. present a global analysis of fresh submarine groundwater discharge (SGD), onshore discharge to the coast (NGD), and saline SGD. They include an analysis relating these groundwater discharges to river discharge, and they consider the potential impact on nutrient flow to coasts. This is an important analysis based on a numerical sensitivity study that involves two-dimensional density-dependent flow and salt transport near the coast.

I have two major concerns with this study.

First, the study includes many novel aspects that could each warrant full article-length discussion: 1) a global analysis of fresh SGD, NGD, and saline SGD (the main text presented here), 2) a new sensitivity study that describes why fresh SGD, NGD, and saline SGD should be high or low in different coastal systems (e.g. Figure 1, S10)—the separation of these three fluxes is itself a relatively novel idea, 3) validation of a new code for variable density flow (e.g. Figures S2 and S3), 4) assessment of potential solute fluxes to the coast (e.g. Figure 4 and Table 1). While the first message is of high-impact (and the last on nutrient inputs), it cannot easily be understood without a mechanistic, process-based explanation of the sensitivity results (#2), which are currently scattered throughout sentences in the main text and supplementary text. Most of the text in L 53-81 is difficult to follow because it depends on the behavior of the sensitivity study, which is never fully described in a mechanistic, process-based way, even in supplementary text. The physics of this problem would be better presented in a full-length journal article that allots ample space for conceptual development, numerical methods, and behavior of the results.

While we agree that the manuscript could also be split up into several separate manuscripts, we do feel that the sensitivity analysis and the global CGD estimates complement each other and that discussing them in one manuscript has its benefits. Thankfully we did have enough space left within the limits set by Nature Communications to add a substantial explanation and discussion of the model sensitivity analysis results. See lines 66 to 136 in the revised manuscript.

Second, the authors suggest that large-scale spatially distributed estimates of fresh SGD have only been published for the US. Another recent study reports estimates for the globe: Zhou, Y. Q., Sawyer, A. H., David, C. H., & Famiglietti, J. S. (2019). Fresh submarine groundwater discharge to the near-global coast. *Geophysical Research Letters*, 46. <https://doi.org/10.1029/2019GL082749>

Many of the results appear to be similar at first glance (for example, concentration of fresh SGD into a small portion of the total coastline, which the authors term “hotspots”, which are often located in tectonically active areas). This study is independent and clearly doesn’t duplicate the Zhou study because Luijendijk et al. use different methods, break groundwater discharge into components of fresh SGD, NGD, and saline SGD, and consider potential nutrient fluxes. However, the results can and should be directly related.

We now discuss the differences in approaches in the revised ms, and discuss the similarities and differences in the introduction (lines 47-54) and in the results section (lines 261-270) of the revised ms. Note that the results by Zhou et al. were published after we had submitted the previous version of the manuscript, and we therefore could not have addressed them in the last version.

Minor suggestions are below:

- L 39: See Zhou et al. (2019)

We have added a more extensive discussion of Zhou et al. (2019) to the new reorganized version of the introduction.

- L 47, 49, and throughout – it took me a while to understand whether “onshore” groundwater discharge and “coastal” groundwater discharge were the same and equivalent to fresh SGD plus NGD (#1 and #2 in L 41-44) or different, and I’m still not sure I have assumed the correct relationships. The authors need to use consistent terminology throughout the main text or define equivalent terms explicitly.

We have reviewed our use of terminology in the manuscript and now consistently use the same four terms, fresh submarine groundwater discharge (fresh SGD), near-shore terrestrial groundwater discharge (NGD), coastal groundwater discharge (CGD = fresh SGD + NGD) and recirculated SGD.

- L 63 and other lines in this paragraph lack context for understanding without a presentation of the sensitivity behavior first

We have expanded the discussion of the sensitivity analysis substantially here.

- L 75: I would disagree that NGD has been overlooked, as it is probably included with fresh SGD in any approach based on a water budget (and is essentially the focus of work by Destouni et al., 2008 on ungauged coastal discharge).

As far as we can tell Destouni et al. 2008 discuss unmonitored near-shore streams and fresh submarine groundwater discharge, but they do not mention near-shore terrestrial groundwater discharge. Even though as a flux NGD may be included implicitly in water budget approaches, we have not found any literature that explicitly mentions near-shore terrestrial groundwater discharge. Instead all the studies that we found assume that the budget term contributes only to fresh submarine groundwater discharge.

- L 89: this observation ties naturally to the results of Michael et al. (2013)—consider citing

Many thanks for pointing out this paper. We now cite Michael in the sensitivity analysis section (line 113-115): “This agrees with results by Michael et al. ¹⁷ that groundwater flow in the majority of the world’s coastline is topography limited instead of recharge limited”

- L 110 and following paragraph—I was unclear on whether NGD is included. Most sentences refer to fresh SGD, but there are also sentences that refer to “terrestrial contributions” that should consider NGD. Even if much NGD is evapotranspired, solutes remain on the landscape, are transported during high tide and flood events to wetlands and estuaries, and exchange with the ocean over different timescales.

Thanks for the suggestion. We agree and now calculate the solute flux using CGD instead of only fresh SGD. We also discuss how the flux would change if only fresh SGD was assumed to transport solutes and we discuss the difference timescales that are likely involved in lines 188-191 of the revised ms.

- L 147: How are groundwater and river fluxes related (over what areas), as rivers are technically point sources to the coast? Is there an integration window for summing up the river fluxes, and if so, what is that coastline length?

The two fluxes are integrated at the watershed scale. We now explain this more thoroughly in the methods section in the manuscript (lines 357 to 366).

- L 169: These results would benefit from comparison with Beusen, which is cited in the Supplementary Text.

The global nitrogen input by fresh SGD in the ocean by Beusen et al. (2013) was already referenced in table 1. We had accidentally omitted the reference from the section on solute input by CGD and have now added this. However, it is very difficult to make a more detailed comparison on the spatial distribution of nitrogen flux / eutrophication risk because the data by Beusen is not accessible to our knowledge. Apart from that we could not find crucial details of their method & results, such as how much groundwater discharge they estimate (only total N flux is reported).

- L 212-213: see Zhou et al. (2019)

We have added a comparison with the recent results by Zhou et al. (2019) here.

In the supplemental material:

- L 151: How deep is most groundwater extraction near the coast? A discussion of how different aquifers are treated in the homogeneous models and where extraction tends to come from might help here.

We assume that most extraction in coastal aquifers is shallow, because extracting from deeper layers would risk hitting the salt water wedge that is located beneath the freshwater part of the aquifer. However, note that groundwater extraction is not actually included in our model simulations, which simulate the natural undisturbed values of coastal groundwater discharge, as clarified in line 246-247 of the manuscript. The data on groundwater extraction are only used for a brief and first order comparison of the natural (undisturbed) CGD that we model with groundwater depletion rates.

- L 169: I could not easily find the description of the landward boundary condition but assume it is specified head.

The landward edge of the model domain is a closed boundary and represents a groundwater divide. We have added an additional line to clarify this.

- L 303: The behavior of the sensitivity analysis needs to be presented mechanistically (how does each component of discharge behave under changes in recharge, permeability, etc, and why) in order to understand the main text.

Agreed, we have added a much more extensive discussion of the sensitivity analysis to the main manuscript, see lines 66 to 136

- L 424: The implications of assuming no rock-water interactions or reactions near the sediment-water interface should be discussed using the marine geochemistry literature.

We have added a new paragraph to section S4 that describes the implications of not assuming water-rock interactions in more detail.

- L 491-494: Most seepage meter studies are conducted in clearly “offshore zones” and capture fresh SGD because otherwise the meters would be exposed during high tide (they are usually deployed for days in advance of measurement to establish steady flow conditions).

While we agree with this in principle, we have opted to keep the comparison between reported fresh SGD and modeled CGD because the partitioning of modeled CGD between onshore (NGD) and offshore (fresh SGD) discharge is highly sensitive to near-shore topographic gradients (as noted in line 164-165 of the main text) and the modelled CGD is a much more robust number. In addition, half the reported studies are not based on seepage meters, but methods such as water and solute budgets for which we cannot exclude a CGD component. We have added a clarification of this point to this section of the SI: “Note that the reported values of fresh SGD were compared to modeled CGD and not modeled fresh SGD because especially for the studies that did not use seepage meters the reported fresh SGD may include discharge above the mean sea level and below the high tide

line, and because the modeled CGD is a more robust number than the contributions of modeled fresh SGD or NGD due to the sensitivity of the partitioning of onshore and offshore discharge to near-shore topographic gradients. If only modelled fresh SGD would be used the values for the modelled fluxes in Fig. S12a would be half or less of the values that are shown.”

References

Destouni, G., F. Hannerz, C. Prieto, J. Jarsjö, and Y. Shibuo (2008), Small unmonitored near-coastal catchment areas yielding large mass loading to the sea, *Global Biogeochem. Cycles*, 22, GB4003, doi:10.1029/2008GB003287.

Michael, H. A., C. J. Russoniello, and L. A. Byron (2013), Global assessment of vulnerability to sea-level rise in topography-limited and recharge-limited coastal groundwater systems, *Water Resour. Res.*, 49, 2228–2240, doi:10.1002/wrcr.20213.

Reviewer #3 (Remarks to the Author):

Review

The flow of fresh groundwater and solutes to the world's oceans and coastal ecosystems
Luijendijk et al.

This manuscript covers an interesting topic to the readers of Nature Communications. The authors show that coastal groundwater discharge can be a significant contributor to the water and chemical budgets of coastal catchments. The simulated flux is highly variable, with hotspots in areas with higher permeabilities, and could result in a high eutrophication risk. This study is the first estimate at the global scale of coastal groundwater discharge and near-shore discharge, therefore includes also information of locations where observations or models are not available.

Although I think this paper is discussing excellent research, I do think methods and results should be more clearly described and discussed to allow the reader to fully understand the importance and novelty of this research. In my comments I focus on a few aspects of the paper where I think a more detailed and/or more clear explanation is needed.

Point 1

The following definitions are introduced in the introduction:

- Submarine groundwater discharge is the flow of fresh or saline groundwater to the oceans.
- Fresh submarine groundwater discharge (fresh SGD) is the fresh component of submarine groundwater discharge .
- Coastal groundwater discharge is the total flux of groundwater to the ocean and can be divided into 1) fresh SGD, 2) near-shore terrestrial groundwater discharge (NGD), 2) recirculating sea water.

The three fluxes, summing up to coastal groundwater discharge are shown in Figure 1. These definitions are clear, also for nonexperts.

In line 47 is written “.. of onshore and offshore groundwater discharge ... coastal discharge”. Are these onshore and offshore fluxes the same as submarine groundwater discharge (fresh or saline) and near-shore terrestrial groundwater discharge? Is ‘coastal groundwater discharge’ meant instead of coastal discharge? (or if not, coastal discharge needs to be explained).

In line 51-53 is written “In addition to submarine groundwater discharge we quantify near-shore terrestrial discharge”. This is confusing as in the previous line it is stated that coastal groundwater discharge was quantified, including fresh SGD, and NGD already. Is this an addition compared to what is estimated in previous studies? Also, is near-shore terrestrial groundwater discharge meant here instead of terrestrial discharge? (if not, near shore terrestrial discharge needs to be explained as well).

Additionally, in Figure 2a different abbreviations are used for fresh SGD and NGD (namely FSGD and NTGD respectively). Abbreviations should be used consistently. Also, the numbering of fluxes in the text is not the same as in the figure (namely numbering in the

figure is 3), 1), 2) if the same numbering as in the text is used). I suggest to make this consistent too. In Figure 2b 'submarine discharge' is used, this should be submarine groundwater discharge (as this is how it is called throughout the text).

Lastly, are the 'density-dependent groundwater models' the same as the series of 'numerical-models of groundwater flow'? And is the 'density-dependent model' (L56) the same as the density-dependent groundwater models?

This are all examples from the introduction part of the paper. I strongly recommend to go through to text and correct these, and similar, inconsistencies in the definitions used.

Reply to the previous comments: We agree and we have clarified the different groundwater discharge components more clearly, have removed all inconsistencies and now consistently use the same terminology throughout the manuscript. The references to groundwater models all referred to the same model code and have now been rephrased as "numerical models of density-dependent groundwater flow"

Also, from this introduction it is not so clear (or convincing) yet what the big selling point is of this research. For example, in this introduction it is not mentioned that this study provides a global estimate. I would suggest to add one or two lines stressing what is new in this study to help the reader understand the added value of this research.

Thanks for the suggestion. We have clarified what is new about this study in lines 60-64 of the introduction

Point 2

Overall it is hard to follow the 'story line' of the paper; it is not always clear what was done and to which purpose and the discussion of the results is not always easy the follow.

For example, the section 'controls on coastal groundwater discharge' start with 'sensitivity analysis...', but what is tested in this sensitivity analysis and how, is not discussed (at least not in the main text). One or two lines briefly stating which parameters were tested, how many runs were done, which data was used etc, would help the reader a lot to interpret and value the results correctly.

We have completely rewritten the results section and have added a much more extensive discussion of the model sensitivity analysis to the manuscript (line 66 to 136).

Also, it is not entirely clear that the examples given in section 'controls on coastal groundwater discharge' are actually two examples of runs done as part as the sensitivity analysis (or did I understand this wrong?).

We have clarified that one of these runs was a separate model run, and the other run was the base case of the model sensitivity analysis in lines 120-124 and 128-131, respectively: “For a separate model run that used the same parameters as the base case of the model sensitivity analysis, but with the global median values of permeability and topographic gradient which were derived from a global geospatial analysis of coastal watersheds (see supplementary information section S1) the modeled coastal groundwater discharge is only 2.0 % and the modelled fresh SGD is only 0.5% of the total recharge volume.”

And

“For the base case model run in our sensitivity analysis, which represents a coastal groundwater system consisting of relatively permeable rocks and a relatively high topographic gradient of 2.5% (Fig. 1b), 50% of the recharge volume contributes to coastal groundwater discharge.
“

In addition to the examples, in lines 56-59 is stated that fresh SGD is insignificant, it might be more logic to first discuss the average scenario where indeed fresh SGD is insignificant and second the exception case where it is 29% of the total recharge flux.

Agreed, the order in which these examples appear in the manuscript now follows this suggestion

Minor comment: in the average example a permeability and gradient are given, in the 29% example only a permeability is given, is the gradient not important?

We now mention the topographic gradient for this model run as well.

As a second example, I did not have any clue what was done to calculate coastal groundwater discharge by paring the results of 351 model runs to geospatial data of 40,082 coastal watersheds (I92-95). From the methods it became clear that linear interpolation was used. I would suggest to rephrases the sentence in the main text to something like this: “.. by linear interpolation of 357 model results to geospatial data of 40,082 coastal watershed, covering information of permeability, topographic gradient, and recharge input.

We followed your suggestion and changed this sentence accordingly.

Minor comment: instead of recharge input you could also use recharge volume, which is more descriptive for recharge input.

Agreed, we changed recharge input to recharge volume.

My general point here is that, although the details of the used methods are hidden in the method sections and Supplementary information, I at least want to understand the basics of each method and step in the research while I am reading the main text.

We have added more text to clarify the methods used in the main manuscript and hope that it is now easier to follow our manuscript.

Another general point: I often have the feeling results of one analysis (e.g sensitivity, or interpolation) can be clustered more together, and then be discussed. For example, in the section 'Global coastal groundwater discharge' the results of the calculation of coastal groundwater discharge, fresh SGD and NGD, are discussed and many percentages and volumes are given scattered over this whole section. Why not structure it a bit more, for example something like this: global coastal groundwater discharge, fresh SGD and NGD, was calculated. Only 4% of recharge is discharged offshore. Of the total SGD flux, 0.06% is fresh SGD, which amounts to 0.2% of the river discharge to oceans. Xx% of recharge is discharged on-shore, which is 0.4% of the global river discharge. And then discuss.

Agreed, we have extensively restructured the results section following this suggestion.

I recommend to go through the text and especially focus on clear writing: make it easy for the reader to follow the story line of this paper.

Minor comments

L30-34: The fresh component of submarine groundwater discharge is critical, due to its high solute and nutrient load³, and has been estimated to be up to 10% of the river discharge to the world's oceans¹ and to equal the inputs by rivers to the for solutes such as carbon⁴, iron⁵, silica^{6,7} and strontium⁸, and potentially buffers ocean acidification with groundwater alkalinity⁹.

ok

In Line 37 "high discharges" is this high submarine discharges or stream discharges or ...

High submarine groundwater discharge, we have clarified this in the manuscript.

L36: "...locally, it is difficult to derive a global estimate from these point estimates,...": which point estimates; are all local information observations? Or also modelling studies (if this is true I am not sure to call it point estimates)

We have rephrased this to clarify that we refer to local estimates here, see lines 44-46. The existing literature on observed values of fresh SGD that we refer to here consist of studies that use seepage meters and local hydrochemical budgets.

L50: "at the global scale"

ok

L53: "coastal water budgets ... " : coastal water and chemical? budgets ...

Changed to coastal water and solute budgets

L57: remove comma before and

Ok

line 64 "... which is lower ..."; this is a bit too vague, give the value or percentage you compare it to. And what do we learn from this; right now, this is presented as a result, but it should be part of a discussion.

The statement on the magnitude of the flux was not meant as a comparison to the percentage of recharge mentioned in the previous sentence, but rather to give the reader a feel for what the magnitude of the flux is, ie. it is so low that it is hardly measurable. We have added a sentence that elaborates this point (line 126-128): "This signifies that in the majority of the world's coastline fresh SGD is expected so low that it is difficult to measure using techniques like seepage meters. "

L74-77: "...but has to our knowledge larger scales": stated on the previous (introduction) section already, the few new information points should be placed in the introduction, can be repeated in the conclusion again.

We have moved this sentence to the introduction.

L77-80: two-dimensional cross-sectional model are the same as the density dependent groundwater models (L47)?

Yes, we changed this sentence to: "Near-shore discharge generates a seepage flux across the land surface that peaks at the coastline and decays exponentially with distance to the coast"

L84: relative to what, the area of watersheds globally?

Added "compared to the watersheds feeding the world's rivers" (line 139-140)

L84-91: repetition of the previous section, not needed.

Agreed, we have removed this section.

L98 “The uncertainly/variability(?) of the estimated global fresh SGD flux”.

We do mean uncertainty here.

L99: which data was used? (add reference)

We have added a reference to this sentence.

L104: add references for the global models.

We added references to the permeability and recharge datasets that were used

L191-193: is it groundwater depletion per year what is meant? And averaged over the globe?

We clarified that this is a spatially resolved estimate of average depletion per year. (line 249)

L223: “...not been explored”. Is this referring the missing observations or (also) missing modelling?

We meant measured and changed the sentence to: “including numerous locations at which coastal groundwater discharge has so far not been measured” (line 291-292)

L226: “serious risk”: high risk

ok

Figure captions of Fig.3 and Fig.4 both miss a brief title that applies to both figure panels. Actually, for figure 3 this title is in the second part “Global maps ...”: move this part up.

We have added new titles to these figure captions.

REVIEWERS' COMMENTS:

Reviewer #1 (Remarks to the Author):

Revision has been made properly according to the reviewer's comments.

Reviewer #3 (Remarks to the Author):

Review

The flow of fresh groundwater and solutes to the world's oceans and coastal ecosystems
Luijendijk et al.

The revised manuscript addressed my main concerns. The paper is now much clearer on used methods/sensitivities/uncertainties and I enjoyed reading the manuscript. I final remark I have concerning the abstract and the percentages given:

L18: ~0.6% of the freshwater input: give the minimum and maximum values as well (thus (0.004-1.3%)) or write approximately 0.004% to 1.3%. Both these options give the estimate range. For the ~2% there is no range, or is there?

L19-21: I was wondering what the freshwater contribution of these regions is (much higher than 0.6%?). Also, I could not find this 20% in the main text. Is this referring to the 26% mentioned on line 234? Also given the estimate range here.

Reviewer #4 (Remarks to the Author):

Major comments:

I was very excited to read this work by Luijendijk and others wherein they present a global analysis to understand how fresh CGD, SGD and NGD (and associated nutrient loads) vary globally by watershed and to develop new estimates of total global fresh CGD (and associated nutrient loads). They relate this to river discharge and consider the impacts on coastlines. It is an important analysis that will impact the way that hydrogeologists and oceanographers approach the study of SGD and the consideration of the associated risks and impacts. I would be excited to see this work in print after revisions.

I have two major concerns with this study:

1) I refrained from reading the previous reviewer comments until I had read the manuscript and my overwhelming takeaway agrees with major criticisms originally presented by reviewers 2 and 3. The information presented seems valuable and novel but I find it difficult to understand the 'story line' and evaluate the work and results because of the density of information and the inevitable space restrictions. For instance, I am still unsure exactly how the models were constructed and how the mechanistic models were upscaled to glean global results even after the rewrite the authors previously conducted (lines 66-136). I agree with R2 that this work would be better presented as at least two (maybe more) independent papers to allow better understanding and evaluation of these findings: e.g. 1) to validate the model and present the sensitivity analysis; 2) to present the global flux and solute results.

2) I agree with R3 (Point 2) and think the authors need to more clearly describe the novelty of this study. I disagree with the assertion that previous studies have not included NGD—Rather, I think a primary strength of this work is that the authors distinguish between SGD and NGD while other large-scale studies report them as a single value. For example, references 11 and 12 inherently

include both NGD and SGD in their budget estimates (as previously noted by R2), and are more accurately reporting CGD than SGD (even though that's not how 11 and 12 report their values). Similarly, several watershed-scale studies have looked at the controls on whether groundwater discharges to upland areas (NGD) or as SGD and discuss how much of the discharge occurs near sea level (e.g. Sanford et al., 2012; Russoniello et al., 2016), so the distinction between NGD and SGD has been previously discussed. Finally, I am not sure that the 2-D steady state models presented here truly distinguish between SGD and NGD—in a system that varied over seasons, storms, and tides, the clear distinction between SGD and NGD would be quite blurred. I think this blurriness requires some additional investigation (or perhaps just some discussion with appropriate citations) to place the distinction in context.

Minor Comments:

I am unconvinced that most NGD would be lost to ET prior to reaching the ocean. If this assertion is going to be made, I feel it requires much stronger evidence, which is likely well beyond the scope of this already information-dense paper. Figure 1 shows that the majority of NGD occurs very close to the shoreline—my experience is that much of this water would discharge to beachfaces and tidal stream banks and almost immediately run downhill to the ocean/estuary and ET would have very little budgetary impact as flowpaths are too short for ET to have much effect. Either way, the solutes carried by NGD would likely reach the ocean whether immediately in a temporally distributed fashion or as solute pulses during high-Q flushing events when solutes delayed by ET were re-dissolved and carried to the sea.

I worry that the conclusions drawn from 2-D steady state simulations (no seasonality or tide) oversimplify the system controls. I would like to see some discussion (or at least some discussion) of how a 3-D approach would have changed the results of this study and a more in-depth discussion of how ignoring transient effects has resulted in a simplified set of results (especially the distinction between SGD and NGD). I can imagine that an undulating coastline could lead to greatly increased CGD over streamflow by presenting steeper topographic gradients that would drive convergent flow into bays.

I appreciate the reviewers work to include the findings of Zhou et al. (2019), in response to the comments of reviewer 2 but think the authors should still compare their findings with Zhou's findings more holistically – for instance discuss more explicitly how the “hot spots” of SGD agree between the two papers. I think this gets back to the significance of this paper that R3 thought was a bit unclear, that the authors are deploying a new more thorough approach to comprehensively validate the findings presented by Zhou et al and to subdivide the fluxes of Zhou et al (CGD) into different buckets (SGD, NGD).

In response to R2's comment: “I would disagree...” L et al wrote: “As far as we can tell...” Me: I think the authors address the explicit comment directly, but miss R2's larger comment, that any water budget approach is inherently representing CGD, not SGD – I think this should be clarified (also see my major comment above)

In response to R2's comment: “L 151: How deep is...” L et al wrote: “We assume most extraction in coastal aquifers is shallow...” Me: I would not have assumed this, especially in the coastal plain aquifers I would most expect to have high rates of SGD. For instance the Atlantic coastal plain (e.g. Gustafson et al., 2019) is one of many locations worldwide (Post et al., 2013) where fresh water is found far offshore of the shoreline. This follows from the framework of Bratton et al. (2010) where these coastal aquifers must be viewed as a system rather than a simple homogeneous system.

Additional minor line-by-line comments:

19-24: The wording of this sentence section makes it unclear if these are findings from this paper

or if these are findings of others.

20-21: Discharge of "nutrients" is high enough to present a eutrophication risk. Suggest changing fresh groundwater to "nutrients" or adding "and nutrients".

22-23: Suggest changing slow rate of flow to "long residence times" (or "travel times") which incorporates velocity and path lengths and is the real risk.

31: Suggest changing load to "loads"

36: I've not seen this term before, and the description is quite vague here "the first hundreds of meters". E.g., vertically or horizontally? I suggest this term be better described at this first use.

39: Suggest changing load to loads, as this refers to more than one nutrient.

40: Recirculated seawater is driven by many additional mechanisms (e.g. Santos et al 2012 in ref list). Suggest adding "mechanisms such as" before "waves" to clarify this.

43: Suggest adding citations after "studies"

55-64: I don't understand the general methods of this study well enough from this paragraph to understand the remaining findings of this paper without reading into the supplement deeply. I don't expect to understand the specifics, but I do expect this paragraph should give me a broad outline so that the supplementary information and methods are not necessary to generally understand the work.

60: Here and elsewhere, I understand the word "significant" to imply a statistical relationship, I suggest choosing a different word or report the statistical significance when using the word significant.

69: I suggest maintaining the acronym NGD once established instead of reverting back to words.

71: What does "modeled seepage" refer to? SGD, NGD, CGD.

71: The phrase "seepage flux" is redundant (here and elsewhere)

75: I'm not sure what "diffusely" means in the context of a 2d model. It appears this comment is in reference to discharge over land surface, but I typically consider diffuse or focused fresh SGD to be brackish or mostly fresh, respectively.

80: I don't think "groundwater flux" can be "transported", but rather that groundwater flows to the coast. Please be more specific in wording here and elsewhere.

84: How do you know this is the reason? Please provide support from sensitivity analysis or citations to prove this point

116: "Groundwater discharge fluxes" is redundant. Also suggest replacing with CGD for consistency with defined acronym.

126: I find this assertion difficult to believe, and at unhelpful at best, because many instances of SGR are reported in literature—SGD and SGR are a continuum. I would imagine some modeling studies have reported very low fresh SGD values, or they could be derived from an analytical model like that in Bokuniewicz (1980).

128: Seepage meters are typically only deployed in settings where high SGD is expected, so I'm

unsure what point is being made here.

129: I do not understand the K-dataset, but I wonder whether it explicitly takes into account the high-K sediments that overly shallow geology in many coastal areas.

141: Though I can infer (I think) the point the authors are trying to make, their logic assumes surface watersheds are equal in size to ground-watersheds (which is seldom true) and that there is no underflow to SGD from inland watersheds, which is also common (e.g. Bratton 2010). The authors should specifically make their point—that this back of the envelope calculation presents a reasonable upper limit!

144: "insignificant"

145: What do these ranges represent? The full range seen in the sensitivity study? Some statistical range of uncertainties?

168: And even smaller, because this recirculated seawater value ignores many of the short-timescale mechanisms (like those presented in Santos et al., 2012) because they operate on timescales much shorter than the time for Radon to reach secular equilibrium.

174-6: See major comment about NGD/SGD/ET above

200: Neat! It would be worth relating this to the work of Zhou et al., 2019, who found similar results. DO hotspots from this paper match in space with those of Zhou? Do the two methods agree?

219-220: The solutes, not the discharge, causes eutrophication and algal blooms. Please clarify in text.

272: Zhou et al is also 3-D, which incorporates geometric considerations beyond the scope of this current study. Worth mentioning the 3-D nature along with the other differences.

292: "Frequently unmonitored": is SGD ever monitored?

311: Neat approach with some very cool new (at least to me) methods and tools. I do find that with so much new information in a single paper it is difficult to understand exactly what was done and how rigorous the work is.

324-5: I wouldn't expect Fresh SGD to be dependent on the saline recirculation at all, but rather the opposite -- terrestrial hydraulic gradient and the fresh seaward groundwater flow generated by that gradient drive the buoyancy-generated recirculation cell. So, instead I would expect the recirculation cell to be dependent on the fresh SGD, a relationship which has been shown by several papers including Smith (2004) and Abarca et al. (2007). I'm not sure what reference 56 is trying to show.

341: Suggest changing "the" to "a" - I imagine more than one such map exists.

383: Or it could suggest that the coastal watershed=surficial watershed assumption is problematic.

References:

Abarca, E., Carrera, J., Sánchez-Vila, X., & Dentz, M. (2007). Anisotropic dispersive Henry problem. *Advances in Water Resources*, 30(4), 913-926.

Bokuniewicz, H. (1980). Groundwater seepage into Great South Bay, New York. *Estuarine and Coastal Marine Science*, 10(4), 437-444.

Bratton, J. F. (2010). The three scales of submarine groundwater flow and discharge across passive continental margins. *The Journal of Geology*, 118(5), 565-575.

Gustafson, C., Key, K., & Evans, R. L. (2019). Aquifer systems extending far offshore on the US Atlantic margin. *Scientific Reports*, 9(1), 8709.

Post, V. E., Groen, J., Kooi, H., Person, M., Ge, S., & Edmunds, W. M. (2013). Offshore fresh groundwater reserves as a global phenomenon. *Nature*, 504(7478), 71-78.

Russoniello, C. J., Konikow, L. F., Kroeger, K. D., Fernandez, C., Andres, A. S., & Michael, H. A. (2016). Hydrogeologic controls on groundwater discharge and nitrogen loads in a coastal watershed. *Journal of Hydrology*, 538, 783-793.

Sanford, W.E., Pope, J.P., Selnick, D.L., and Stumvoll, R.F., 2012, Simulation of groundwater flow in the shallow aquifer system of the Delmarva Peninsula, Maryland and Delaware: U.S. Geological Survey Open-File Report 2012-1140, 58 p.

Smith, A. J. (2004). Mixed convection and density-dependent seawater circulation in coastal aquifers. *Water Resources Research*, 40(8).

Reviewer #3 (Remarks to the Author):

Review

The flow of fresh groundwater and solutes to the world's oceans and coastal ecosystems
Luijendijk et al.

The revised manuscript addressed my main concerns. The paper is now much clearer on used methods/sensitivities/uncertainties and I enjoyed reading the manuscript. I final remark I have concerning the abstract and the percentages given:

L18: ~0.6% of the freshwater input: give the minimum and maximum values as well (thus (0.004-1.3%)) or write approximately 0.004% to 1.3%. Both these options give the estimate range. For the ~2% there is no range, or is there?

Response: Agreed, we now mention the range in the abstract. For the ~2% solute flux we now mention the minimum and maximum value of the solute flux of 0.003%-7.7% which is based on the range of values listed in Table 1.

L19-21: I was wondering what the freshwater contribution of these regions is (much higher than 0.6%?). Also, I could not find this 20% in the main text. Is this referring to the 26% mentioned on line 234? Also given the estimate range here.

R: We now mention the correct numbers and the range of the fluxes to the three ecosystems in the abstract.

Reviewer #4 (Remarks to the Author):

Major comments:

I was very excited to read this work by Lujendijk and others wherein they present a global analysis to understand how fresh CGD, SGD and NGD (and associated nutrient loads) vary globally by watershed and to develop new estimates of total global fresh CGD (and associated nutrient loads). They relate this to river discharge and consider the impacts on coastlines. It is an important analysis that will impact the way that hydrogeologists and oceanographers approach the study of SGD and the consideration of the associated risks and impacts. I would be excited to see this work in print after revisions.

I have two major concerns with this study:

1) I refrained from reading the previous reviewer comments until I had read the manuscript and my overwhelming takeaway agrees with major criticisms originally presented by reviewers 2 and 3. The information presented seems valuable and novel but I find it difficult to understand the ‘story line’ and evaluate the work and results because of the density of information and the inevitable space restrictions.

For instance, I am still unsure exactly how the models were constructed and how the mechanistic models were upscaled to glean global results even after the rewrite the authors previously conducted (lines 66-136).

R: We agree that the methods used to derive the global estimates were still not that well described in the previous version of the manuscript and we have added additional text to clarify how the model results were upscaled to calculate the global estimates of the fluxes and we have improved the figure caption for figure 2

I agree with R2 that this work would be better presented as at least two (maybe more) independent papers to allow better understanding and evaluation of these findings: e.g. 1) to validate the model and present the sensitivity analysis; 2) to present the global flux and solute results.

R: While we agree that the work could also be split into two papers, we still feel that the sensitivity analysis provides essential background to the global estimate. We also have competition from other research groups on the global estimates, so we prefer to get this work out sooner than later.

2) I agree with R3 (Point 2) and think the authors need to more clearly describe the novelty of this study. (R3 point 2: “Overall it is hard to follow the ‘story line’ of the paper; it is not always clear what was done and to which purpose and the discussion of the results is not always easy to follow.”)

R: We have rewritten the manuscript extensively following the remarks in the previous review round and specifically R3 point 2. We have reorganized the sensitivity analysis and results section completely and are not entirely sure how to respond to this new comment. It would be helpful if R4 would suggest more specific parts where the storyline is unclear.

The novelty of the paper is clearly described in the following line of the introduction, i.e., it is the second global estimate of these fluxes and the first that explicitly takes groundwater flow processes and the partitioning of terrestrial and submarine discharge into account:

“Our study is a significant advance on recently published large-scale estimates of fresh SGD^{11,12} that are based on surface water budgets of coastal watersheds, because it explicitly takes into account groundwater flow processes in the subsurface to resolve all three fluxes of groundwater to the ocean, including density-dependent flow that is critical for resolving submarine and terrestrial coastal groundwater fluxes, and because we use the best available global distributed input data”

I disagree with the assertion that previous studies have not included NGD—Rather, I think a primary strength of this work is that the authors distinguish between SGD and NGD while other large-scale studies report them as a single value. For example, references 11 and 12 inherently include both NGD and SGD in their budget estimates (as previously noted by R2), and are more accurately reporting CGD than SGD (even though that's not how 11 and 12 report their values). Similarly, several watershed-scale studies have looked at the controls on whether groundwater discharges to upland areas (NGD) or as SGD and discuss how much of the discharge occurs near sea level (e.g. Sanford et al., 2012; Russoniello et al., 2016), so the distinction between NGD and SGD has been previously discussed.

R: We agree that NGD may have implicitly been included in some previous studies. However to our knowledge it has not been reported as a distinct flux, although these are otherwise awesome studies even Sanford 2012 and Russoniello et al (2016) do not report that discharge occurs both above and below the shoreline. For modelling studies like Sanford et al. (2012) this is simply a consequence of the relatively high resolution needed to distinguish these two fluxes, Sanford et al use a grid size of 160 m, which in our models would cover most of the area where NGD and fresh SGD occurs. At any rate Sanford et al. (2012) report only terrestrial discharge to streams and submarine groundwater discharge, not NGD. Similarly Russoniello et al (2016) equate groundwater discharge to bays (which presumably includes NGD as well) with SGD and do not distinguish terrestrial and submarine discharge at the shoreline. References 11 and 12 (Zhou et al. 2019 and Sawyer et al. 2016) definitely report their values as fresh SGD, and do not mention any possibility of the terrestrial discharge (NGD) of part of the water budget term that is interpreted as fresh SGD. We therefore feel justified in our statement that this flux has been overlooked in large scale analysis of groundwater fluxes (“Our analysis shows that coastal discharge is subdivided in fresh submarine groundwater discharge and a roughly equally important component of terrestrial near-shore discharge, which has been overlooked in most previous analyses.”), or that previous studies have not reported this flux as stated in the introduction (“In addition, previous large-scale estimates have not reported the near-shore terrestrial discharge of groundwater that may affect coastal water and solute budgets, evapotranspiration and ecosystems.”). We did add a sentence to state that some previous estimates may have implicitly included NGD in fresh SGD estimates: “, which has been overlooked in most previous analyses, and has instead been lumped with fresh SGD in water budget analyses and model studies.”

Finally, I am not sure that the 2-D steady state models presented here truly distinguish between SGD and NGD—in a system that varied over seasons, storms, and tides, the clear distinction between SGD and NGD would be quite blurred. I think this blurriness requires some additional investigation (or perhaps just some discussion with appropriate citations) to place the distinction in context.

R: We agree that the boundary between SGD and NGD varies over time, which is not captured by our model experiments that model a steady-state system. We also agree that it would be worthwhile to investigate this further using for instance numerical models with transient boundary conditions that reflect the effects of waves and tides. However we feel that this would be more suited for a follow up publication.

Minor Comments:

I am unconvinced that most NGD would be lost to ET prior to reaching the ocean. If this assertion is going to be made, I feel it requires much stronger evidence, which is likely well beyond the scope of this already information-dense paper. Figure 1 shows that the majority of NGD occurs very close to the shoreline—my experience is that much of this water would discharge to beachfaces and tidal stream banks and almost immediately run downhill to the ocean/estuary and ET would have very little budgetary impact as flowpaths are too short for ET to have much effect. Either way, the solutes carried by NGD would likely reach the ocean whether immediately in a temporally distributed fashion or as solute pulses during high-Q flushing events when solutes delayed by ET were re-dissolved and carried to the sea.

R: We agree and have rephrased the sentence that discusses this in the manuscript (“In most cases NGD does not exceed potential evapotranspiration rates. However, NGD exceeds potential evapotranspiration²⁷ in 28 (0.07-58) % of the global coastline where it contributes to surface runoff and baseflow close to the shoreline.”). The reason for stating this is that in most cases the discharge flux is lower than the ET flux, which means that in theory ET could keep up with the groundwater discharge. However we did not mean to claim that we have accurate information on the exact discharge mechanism of ET and a more in depth exploration of this would in our opinion best be left for follow up studies.

I worry that the conclusions drawn from 2-D steady state simulations (no seasonality or tide) oversimplify the system controls. I would like to see some discussion (or at least some discussion) of how a 3-D approach would have changed the results of this study and a more in-depth discussion of how ignoring transient effects has resulted in a simplified set of results (especially the distinction between SGD and NGD). I can imagine that an undulating coastline could lead to greatly increased CGD over streamflow by presenting steeper topographic gradients that would drive convergent flow into bays.

R: To what extent 2D cross-sectional models of groundwater discharge are representative of the 3D reality is extensively discussed and explored with model experiments in the supplementary information (see Supplementary Note 2). The effect of the steady-state assumption has to been explored in a previous study that we cite in the manuscript and that found that while transient flow caused by wave setup and tides was not surprisingly very important in controlling the flux of recirculated SGD, the fresh water discharge was relatively insensitive to transient effects (Li et al., 1999). Therefore, while we agree that our model makes a large number of necessary simplifications, our first order conclusions are still relatively robust.

I appreciate the reviewers work to include the findings of Zhou et al. (2019), in response to the comments of reviewer 2 but think the authors should still compare their findings with Zhou’s findings more holistically – for instance discuss more explicitly how the “hot spots” of SGD agree between the two papers. I think this gets back to the significance of this paper that R3 thought was a bit unclear, that the authors are deploying a new more thorough approach to comprehensively validate the findings presented by Zhou et al and to subdivide the fluxes of Zhou et al (CGD) into different buckets (SGD, NGD).

R: We disagree here, our method is very different from the method employed by Zhou et al. (1999), which we find to be rather shaky and which to our knowledge has not been validated yet with any

field data, physical or model experiments. We therefore consider it somewhat pointless to “validate” the estimates by Zhou et al. or to somehow use our model results to partition the total discharge fluxes reported by Zhou et al. into terrestrial and submarine discharge. We also feel that at any rate the manuscript is information dense enough and introducing another detailed comparison would make the manuscript somewhat hard to follow for most readers.

In response to R2’s comment: “I would disagree...” L et al wrote: “As far as we can tell...” Me: I think the authors address the explicit comment directly, but miss R2’s larger comment, that any water budget approach is inherently representing CGD, not SGD – I think this should be clarified (also see my major comment above)

(- L 75: I would disagree that NGD has been overlooked, as it is probably included with fresh SGD in any approach based on a water budget (and is essentially the focus of work by Destouni et al., 2008 on ungauged coastal discharge).”

R: This discussion is a bit semantic. We agree that water budget approaches really quantify CGD, not SGD. But if these studies only report this as SGD I think we are right in pointing out that these estimates include terrestrial discharge fluxes as well. Also see our response to a similar comment above.

In response to R2’s comment: “L 151: How deep is...” L et al wrote: “We assume most extraction in coastal aquifers is shallow...” Me: I would not have assumed this, especially in the coastal plain aquifers I would most expect to have high rates of SGD. For instance the Atlantic coastal plain (e.g. Gustafson et al., 2019) is one of many locations worldwide (Post et al., 2013) where fresh water is found far offshore of the shoreline. This follows from the framework of Bratton et al. (2010) where these coastal aquifers must be viewed as a system rather than a simple homogeneous system. *(“- L 151: How deep is most groundwater extraction near the coast? A discussion of how different aquifers are treated in the homogeneous models and where extraction tends to come from might help here.”)*

R: We are still pretty sure that the large majority of wells in coastal aquifer systems are shallow. The existence deep offshore freshwater stores has only become apparent after the publication by Post (2013), the earlier local studies that Post summarizes were not widely known or cited in our experience. To our knowledge this publication has not yet led to new deep wells near the coast or at sea to tap this new resource. We have discussed this briefly at conferences with Vincent Post and Mark Person, who was a co-author on the 2013 paper. At any rate, as also stated in our previous answer the question of how deep these wells are is a bit irrelevant for our simple first order comparison of groundwater depletion and natural discharge rates. See previous reply: *“However, note that groundwater extraction is not actually included in our model simulations, which simulate the natural undisturbed values of coastal groundwater discharge, as clarified in line 246-247 of the manuscript. The data on groundwater extraction are only used for a brief and first order comparison of the natural (undisturbed) CGD that we model with groundwater depletion rates.”*

Additional minor line-by-line comments:

19-24: The wording of this sentence section makes it unclear if these are findings from this paper or if these are findings of others.

R: We have rephrased the abstract and hope that this is now clearer.

20-21: Discharge of “nutrients” is high enough to present a eutrophication risk. Suggest changing fresh groundwater to “nutrients” or adding “and nutrients”.

R: Agreed

22-23: Suggest changing slow rate of flow to “long residence times” (or "travel times") which incorporates velocity and path lengths and is the real risk.

R: This sentence was removed from the abstract

31: Suggest changing load to “loads”

R: Agreed

36: I’ve not seen this term before, and the description is quite vague here “the first hundreds of meters”. E.g., vertically or horizontally? I suggest this term be better described at this first use.

R: We have removed the statement on hundreds of meters that made this definition somewhat vague.

39: Suggest changing load to loads, as this refers to more than one nutrient.

R: Agreed

40: Recirculated seawater is driven by many additional mechanisms (e.g. Santos et al 2012 in ref list). Suggest adding “mechanisms such as” before “waves” to clarify this.

R: Agreed

43: Suggest adding citations after “studies”

R: We would have liked to add citations here but are already over our max. citation limit for the manuscript.

55-64: I don’t understand the general methods of this study well enough from this paragraph to understand the remaining findings of this paper without reading into the supplement deeply. I don't expect to understand the specifics, but I do expect this paragraph should give me a broad outline so that the supplementary information and methods are not necessary to generally understand the work.

R: This sentence was meant to introduce the methods and novelty of our study, not discuss the methods in detail. We prefer to provide more detail in the individual sections controls on coastal

groundwater discharge and global coastal groundwater discharge. We have added the following sentences to these sections to clarify the methods:

“We modeled coastal groundwater flow using a numerical model in which the rate of groundwater discharges both on land and at the seafloor and the rate of discharge was allowed to vary freely (see Methods for a more detailed description of the model approach). The model experiments are based on a conceptual model shown in Fig. 1a. The modelled groundwater flow paths, salinity and discharge rates for a typical model setup are shown in Figure 1b”

“Model sensitivity analysis that explores the response of groundwater discharge to variation in groundwater recharge, size of contributing area, permeability and topographic gradient show that”

And on the global scale estimate:

“For each of the world’s coastal watersheds the CGD, NGD and fresh SGD fluxes were calculated by linear interpolation of the CGD, NGD and fresh SGD fluxes in the model runs with the values of permeability, recharge volume and topographic gradient that were closest to the values of each watershed (see Methods).”

60: Here and elsewhere, I understand the word “significant” to imply a statistical relationship, I suggest choosing a different word or report the statistical significance when using the word significant.

R: We would like to leave this issue to the editor. In our experience the words significant or insignificant can also be used in a non-statistical sense. See for instant the oxford dictionary for significant (<https://www.oed.com/view/Entry/179569?redirectedFrom=significant#eid>) where one of the meanings is given as “Sufficiently great or important to be worthy of attention; noteworthy; consequential, influential.”

69: I suggest maintaining the acronym NGD once established instead of reverting back to words.

R: Agreed.

71: What does “modeled seepage” refer to? SGD, NGD, CGD.

R: NGD, we have rephrased this accordingly.

71: The phrase “seepage flux” is redundant (here and elsewhere)

R: Agreed, we have rewritten this sentence.

75: I’m not sure what “diffusely” means in the context of a 2d model. It appears this comment is in reference to discharge over land surface, but I typically consider diffuse or focused fresh SGD to be brackish or mostly fresh, respectively.

R: We do mean discharge over the land surface here and have removed the part of the sentence that refers to diffuse discharge to avoid confusion.

80: I don't think "groundwater flux" can be "transported", but rather that groundwater flows to the coast. Please be more specific in wording here and elsewhere.

R: Agreed, we have rephrased this to groundwater flow instead.

84: How do you know this is the reason? Please provide support from sensitivity analysis or citations to prove this point

R: This is basic physics, we are not sure how to add anything to this statement. That dense groundwater is more difficult to displace than fresh water follows from the definition of groundwater potential by Hubbert (1940). One of the authors has also co-authored a separate paper on this:

<https://agupubs.onlinelibrary.wiley.com/doi/full/10.1029/2018GL078409>, although this study focused on deep fluid flow in basins and not coastal groundwater flow.

116: "Groundwater discharge fluxes" is redundant. Also suggest replacing with CGD for consistency with defined acronym.

R: Agreed, we have rephrased this sentence.

126: I find this assertion difficult to believe, and at unhelpful at best, because many instances of SGR are reported in literature—SGD and SGR are a continuum. I would imagine some modeling studies have reported very low fresh SGD values, or they could be derived from an analytical model like that in Bokuniewicz (1980).

R: We meant to refer to measured fresh SGD values, not modelled values. We have rephrased this to "which is lower than the lowest measured value of fresh SGD in the literature known to us"¹⁸

128: Seepage meters are typically only deployed in settings where high SGD is expected, so I'm unsure what point is being made here.

R: The point being made here is that it will be difficult to measure fresh SGD in most of the world's coastline, and therefore extrapolating local measurements to regional or global estimates will lead to strongly biased values.

129: I do not understand the K-dataset, but I wonder whether it explicitly takes into account the high-K sediments that overly shallow geology in many coastal areas.

R: In principle it does, the permeability map is based on surface geology. The degree to which this does depends on the accuracy of local lithological maps that the global lithology and permeability maps are based on. We do acknowledge the limitations of this global dataset in several places in the manuscript.

141: Though I can infer (I think) the point the authors are trying to make, their logic assumes surface watersheds are equal in size to ground-watersheds (which is seldom

true) and that there is no underflow to SGD from inland watersheds, which is also common (e.g. Bratton 2010). The authors should specifically make their point—that this back of the envelope calculation presents a reasonable upper limit!

R: We have added a sentence to clarify the point we try to make here:

“However, as discussed in the previous section, for most coastal watersheds CGD is only a fraction of the total recharge volume.”

144: “insignificant”

R: See response to previous comment

145: What do these ranges represent? The full range seen in the sensitivity study? Some statistical range of uncertainties?

R: We have added a sentence to clarify how the uncertainty ranges were calculated:

“The uncertainty ranges reported here represent the CGD, NGD and fresh SGD fluxes using end-member estimates for permeability, groundwater recharge, contributing area and topographic gradients.”

168: And even smaller, because this recirculated seawater value ignores many of the short-timescale mechanisms (like those presented in Santos et al., 2012) because they operate on timescales much shorter than the time for Radon to reach secular equilibrium.

R: While we find this point convincing, we do prefer to keep this statement as is and to keep our reference to a published estimate of total SGD here, instead of opening up a new discussion in the manuscript on a previously published estimate. This point is also not central to any of our conclusions.

174-6: See major comment about NGD/SGD/ET above

R: We have rephrased this sentence to not make a strong claim on the link between NGD and ET: “In most cases NGD does not exceed potential evapotranspiration rates. However, NGD exceeds potential evapotranspiration²⁷ in 28 (0.07-58) % of the global coastline where it contributes to surface runoff and baseflow close to the shoreline.”

200: Neat! It would be worth relating this to the work of Zhou et al., 2019, who found similar results. DO hotspots from this paper match in space with those of Zhou? Do the two methods agree?

R: We do compare our results to Zhou et al in a separate section (Comparison with published large-scale models of coastal groundwater discharge). We are however somewhat skeptical of Zhou et al.’s methods. We have not seen any validation or justification of why their water budget approach should work or not in Zhou et al or Sawyer (2016) where this method was first introduced. Therefore we consider our results not compatible, and would prefer to not compare the locations of hotspots in detail.

219-220: The solutes, not the discharge, causes eutrophication and algal blooms. Please clarify in text.

R: agreed, we have rephrased this sentence accordingly:

“Coastal groundwater discharge can control the salinity, nutrient budget and productivity of coastal lagoons³¹, salt marshes³² and mangroves³³, and the associated solute flux can cause eutrophication³⁴, algal blooms³⁵ and the degradation of coral reefs³⁶”

272: Zhou et al is also 3-D, which incorporates geometric considerations beyond the scope of this current study. Worth mentioning the 3-D nature along with the other differences.

R: Our reading of the Zhou et al approach used for their near global estimate is that it is definitely a map-view 2D, water-budget based approach, there is no 3D aspect in there as far as we can tell.

292: “Frequently unmonitored”: is SGD ever monitored?

R: Not as far as we are aware. We have rephrased this sentence.

311: Neat approach with some very cool new (at least to me) methods and tools. I do find that with so much new information in a single paper it is difficult to understand exactly what was done and how rigorous the work is.

R: We have moved the method description in the SI to the main manuscript and have also clarified the methods to upscale the model experiments to a global estimate of CGD, as described in an earlier response. We feel that our approach of using large sets of 2D models to quantify large scale fluxes is new and somewhat unconventional, but the individual components of this workflow are all relatively well established. The new density-driven model code is well validated and based on a well-established model code (escript). We therefore hope that in the new version it is somewhat easier to follow what was done, we are not entirely sure otherwise what to improve in the text to make this clearer.

324-5: I wouldn't expect Fresh SGD to be dependent on the saline recirculation at all, but rather the opposite -- terrestrial hydraulic gradient and the fresh seaward groundwater flow generated by that gradient drive the buoyancy-generated recirculation cell. So, instead I would expect the recirculation cell to be dependent on the fresh SGD, a relationship which has been shown by several papers including Smith (2004) and Abarca et al. (2007). I'm not sure what reference 56 is trying to show.

R: Reference 56 shows that fresh submarine groundwater discharge is relatively insensitive to transient effects such as wave setup and tides. We agree that fresh water discharge influences circulation to some extent and not the other way around. We have rephrased this sentence accordingly to avoid confusion.

“Note that the submarine discharge of fresh groundwater is relatively insensitive to transient flow induced by wave set-up and tides⁵⁶”

341: Suggest changing “the” to “a” – I imagine more than one such map exists.

R: Agreed

383: Or it could suggest that the coastal watershed=surficial watershed assumption is problematic.

R: This is true in principle. However as explained in the SI all of the reported values are located in watersheds with perennial streams that should also channel substantial parts of the overall groundwater recharge. Therefore even if one would assume substantially larger contributing areas than coastal watersheds it would still be difficult to reconcile the reported fresh SGD values with the onshore water budgets. We have clarified this in the main manuscript as well:
“This in spite of the fact that all of the reported values are located in watersheds with perennial streams that also discharge significant parts of the overall groundwater recharge.”

References:

Abarca, E., Carrera, J., Sánchez-Vila, X., & Dentz, M. (2007). Anisotropic dispersive Henry problem. *Advances in Water Resources*, 30(4), 913-926.

Bokuniewicz, H. (1980). Groundwater seepage into Great South Bay, New York. *Estuarine and Coastal Marine Science*, 10(4), 437-444.

Bratton, J. F. (2010). The three scales of submarine groundwater flow and discharge across passive continental margins. *The Journal of Geology*, 118(5), 565-575.

Gustafson, C., Key, K., & Evans, R. L. (2019). Aquifer systems extending far offshore on the US Atlantic margin. *Scientific Reports*, 9(1), 8709.

Post, V. E., Groen, J., Kooi, H., Person, M., Ge, S., & Edmunds, W. M. (2013). Offshore fresh groundwater reserves as a global phenomenon. *Nature*, 504(7478), 71-78.

Russoniello, C. J., Konikow, L. F., Kroeger, K. D., Fernandez, C., Andres, A. S., & Michael, H. A. (2016). Hydrogeologic controls on groundwater discharge and nitrogen loads in a coastal watershed. *Journal of Hydrology*, 538, 783-793.

Sanford, W.E., Pope, J.P., Selnick, D.L., and Stumvoll, R.F., 2012, Simulation of groundwater flow in the shallow aquifer system of the Delmarva Peninsula, Maryland and Delaware: U.S. Geological Survey Open-File Report 2012–1140, 58 p.

Smith, A. J. (2004). Mixed convection and density-dependent seawater circulation in coastal aquifers. *Water Resources Research*, 40(8).

References

Li, L., Barry, D.A., Stagnitti, F., Parlange, J.Y., 1999. Submarine groundwater discharge and associated chemical input to a coastal sea. *Water Resour. Res.* 35, 3253–3259.
doi:10.1029/1999WR900189